# Cytoreductive surgery and hyperthermic intrathoracic chemotherapy in thymic epithelial tumors with pleural spread or recurrence: a prospective, single-arm, phase II study

Shuai Wang [1,2,4], Xinyu Yang [1,2,4], Jiahao Jiang [1,2,4], Fei Liang [3], Yuansheng Zheng[1,2], Yongqiang Ao [1,2], Jian Gao [1,2], Hao Wang[1,2], Lijie Tan[1,2] ✉ & Jianyong Ding [1,2] ✉

Pleural spread or recurrence of thymic epithelial tumors (TETs) is a tricky puzzle in the clinic and there is currently no recognized effective treatment. This trial evaluated the safety and efficacy of cytoreductive surgery and hyperthermic intrathoracic chemotherapy (S-HITOC) for TETs with pleural spread or recurrence. Here, we reported short-term outcomes of enrolled 45 patients receiving S-HITOC with 25 mg/m$^2$ doxorubicin and 50 mg/m$^2$ cisplatin. The pleural tumor index (PTI) has been proposed for evaluating pleural tumor burden. Treatment-related adverse events of grade ≥3 occurred in eight (17.8%) patients. The pain Visual analog scale (VAS) score was 5.4 ± 1.9 on the 1st day after treatment and was similar to that at baseline level on the 7th day after treatment ($p = 0.218$). There was no significant difference in the quality of life score ($p = 0.676$) between baseline and the 60th day after treatment. The estimated 2-year PFS and OS rates were 82.8% and 100.0%, respectively. Subgroup analyses revealed that patients with PTI scores >10 had worse PFS than those with PTI scores ≤10 ($p < 0.001$). S-HITOC had a manageable complication rate. Early clinical outcomes confirmed that S-HITOC offers encouraging oncological benefits for TETs and satisfactory control of myasthenia gravis. Trial number: NCT05446935.

Thymic epithelial tumors (TETs) are the most common malignancies in the anterior mediastinum. Both pleural and pericardial dissemination may be detected as an initial presentation of stage IVA disease (according to the Masaoka and TNM staging systems), which accounts for less than 10% of all cases or during surveillance at the diagnosis of recurrence in up to 30% of radically resected TETs[1–3]. The treatments of TETs with pleural dissemination (TPD), including de novo Masaoka stage IVA TETs (DNT) and TETs with pleural recurrence (TPR), remain challenging. Surgery alone has been proven ineffective in achieving efficient local control, with complete resection reported in less than

[1]Department of Thoracic Surgery, Zhongshan Hospital, Fudan University, Shanghai 200032, China. [2]Cancer Center, Zhongshan Hospital, Fudan University, Shanghai 200032, China. [3]Department of Biostatistics, Zhongshan Hospital, Fudan University, Shanghai, China. [4]These authors contributed equally: Shuai Wang, Xinyu Yang, Jiahao Jiang. ✉e-mail: tan.lijie@zs-hospital.sh.cn; ding.jianyong@zs-hospital.sh.cn

two-thirds of cases, even in cases of extended resection[4,5]. While there is a limited survival benefit from chemotherapy and/or radiotherapy, which is often associated with considerable morbidity and mortality[6].

Cytoreductive surgery followed by hyperthermic intrathoracic chemotherapy (S-HITOC) is an alternative treatment for clearing residual tumors[7]. HITOC has the potential to increase local disease control, prolong disease-free intervals, prevent local recurrence, and eventually improve survival in patients with TPR or DNT. The German HITOC study confirmed that a 5-year disease-free survival (DFS) rate of 57% and a 5-year overall survival (OS) rate of 94% were achieved by S-HITOC in thymoma patients with pleural metastasis[8]. Notably, even in patients with incomplete tumor resection, a 5-year OS rate of 51% was achieved. A multicenter retrospective study conducted in France revealed that S-HITOC increased the survival rate, with a hospital mortality rate of 2.5% and a specific HITOC complication rate of 10% among DNT thymomas[9]. In addition, Aprile et al. reported that S-HITOC was associated with longer DFS than surgery alone in TPR patients[10]. However, the limitations of these studies are obvious, including their retrospective nature and the heterogeneity of treatment regimens, which limit the development of a standard therapeutic regimen for S-HITOC.

In this work, the CHOICE study (ClinicalTrials.gov identifier: NCT05446935) was conducted to explore the safety and efficacy of a uniform S-HITOC mode for TETs with pleural dissemination. We enrolled eligible patients according to rigorous criteria under the multidisciplinary model. Furthermore, a scoring system is developed to evaluate pleural tumor burden and distribution. This is the first prospective study to report the perioperative outcomes and short-term survival of TETs patients with S-HITOC.

## Results

### Participants

Between August 1, 2021, and February 29, 2024, 45 patients received S-HITOC at Zhongshan Hospital of Fudan University (Supplement Figs. 1–2). The participant characteristics were shown in Table 1. Among the patients, 22 (48.9%) were male, and 7 (15.6%) were older than 65 years. Nineteen (42.2%) patients had comorbidities, and seven (15.6%) patients had a Charlson Comorbidity Index (CCI) of >3. Six patients were thymic carcinoma. Among 39 thymoma patients, 1 (2.2%), 1 (2.2%), 3 (6.6%), 14 (31.1%), 10 (22.2%) and 10 (22.2%) patients were type A, AB, B1, B2, B2/B3 mix, and B3, respectively (Table 1). Pretreatment myasthenia gravis (MG) was observed in 12 (26.7%) patients, of which 6 (13.3%), 4 (8.9%), 1 (2.2%), and 1 (2.2%) were classified as the Myasthenia Gravis Foundation of America (MGFA) class I, II, III, and IV, respectively. Twelve (26.7%) patients were diagnosed with DNT. TPR occurred in 33 (73.3%) patients, among whom 16 (35.6%) exhibited recurrence more than five years after surgery. Forty (88.9%) patients had pathological stage IVA disease, and 6 (13.3%) patients had nodal metastasis. Preoperative therapies were administered in 30 patients (66.7%), including radiotherapy, chemotherapy, or chemoradiotherapy (Supplement Table 1).

### Surgical outcomes

The perioperative outcomes were summarized in Table 2. All patients completed the planned cytoreductive surgery and HITOC. The pleural tumor burden was assessed by dividing the pleural cavity into five zones and the pleural tumor index (PTI) was proposed as a reproducible index for evaluating the pleural tumor burden and distribution (Fig. 1). Forty-one (91.1%), 34 (75.6%), 23 (51.1%), 32 (71.1%), and 19 (42.2%) patients experienced tumor metastasis in Zones I, II, III, IV and V, respectively. Typical metastatic nodules and resections were shown in Fig. 2 and Supplement Figs. 3 and 4. The median PTI [IQR] score was 6 [5,9]. Three (6.7%) patients received pleurotomy/decortication (P/D) and the others (93.3%) received extended pleurectomy/decortication (eP/D). Among the 45 patients, 37 (82.2%), 35 (77.8%), 7 (15.6%), and 1

(2.2%) patient(s) underwent partial diaphragmatic resection and repair, partial lung resection, partial pericardiectomy and repair, and azygos vein resection, respectively. These resection procedures resulted in 23 cases (51.1%) of complete cytoreductive surgery without residual visible disease (R0), eight cases (17.8%) of optimal cytoreductive surgery with residual tumors measuring no more than 10 mm (R1), and 14 cases (31.1%) of incomplete cytoreductive surgery with residual lesions measuring >10 mm in diameter (R2).

### Treatment-related adverse events

The treatment-related adverse events were shown in Table 3. The five most common treatment-related adverse events were pleural effusion ($n = 13$, 28.2%), increased AST/ALT ($n = 11$, 24.4%), anemia ($n = 10$, 22.2%), fever ($n = 9$, 20.0%), and pneumothorax ($n = 9$, 20.0%). Most of these events were grade 1 or 2. Pulmonary treatment-related complications reported in ≥5% of the patients were pleural effusion (28.2%), pneumothorax (20.0%), subcutaneous emphysema (11.1%), and pneumonia (8.7%). Prolonged QT (according to electrocardiography) ($n = 5$; 11.1%) was the most common cardiac complication. The serum creatinine level increased, and renal function abnormalities occurred in 4 (8.9%) and 1 (2.2%) patient(s), all of which were grade 1–2. Grade 3a treatment-related adverse events occurred in eight (17.8%) patients, including pleural effusion (4.4%), pneumothorax (4.4%), hemothorax (2.2%), atrial fibrillation (2.2%), and intolerable pain (2.2%). No reoperation was performed in the patients who experienced treatment-related adverse events. All patients were discharged within 14 days of treatment, and the median length of posttreatment stay (LOS) was 4.0 days.

### Posttreatment pain and quality of life (QoL) assessment

All the patients completed the planned posttreatment pain and QoL assessments. The pain Visual analog scale (VAS) score was $5.4 \pm 1.9$ on the 1st day after treatment and then decreased to $3.1 \pm 1.8$ on the 3rd day after treatment. Finally, the score on the 7th day after treatment was similar to that at baseline ($0.6 \pm 0.9$ vs. $0.4 \pm 0.6$, $p = 0.218$) (Supplement Fig. 5). In addition, QoL recovery was observed from treatment to postoperative day 60, and there was no significant difference in the QoL score ($77.1 \pm 9.2$ vs. $76.3 \pm 8.9$, $p = 0.676$) between baseline and the 60th day after treatment (Supplement Fig. 6).

Among the 12 patients with preoperative MG, all patients (100%) were remission at the latest visit, including three (25.0%) patients with complete stable remission (CSR), six (50.0%) patients with pharmacological remission (PR), and three (25.0%) patients with minimal manifestations (MM) (Supplement Fig. 7). The remission rate is 100%. Notably, the mean glucocorticoid usage (pretreatment vs. posttreatment: $20.8 \pm 11.2$ vs. $10.0 \pm 7.4$ mg/d, $p = 0.011$) was reduced at the 1 month after S-HITOC in all these patients (Supplement Fig. 8).

### Survival

The median [IQR] follow-up was 12.2 [8.1, 20.2] months at the latest visit. Among the 43 surviving patients, 22 (51.2%) completed the posttreatment 1-year follow-up and 10 (23.2%) completed the posttreatment 2-year follow-up. At the latest visit, five patients (11.1%) experienced posttreatment recurrence, including two patients with pleural metastasis, one patient with lung and bone metastasis, one patient with pericardiac and pleural metastasis, and one patient with lung metastasis. The 1- and 2-year progression-free survival (PFS) rates were 97.3% (92.2–100.0%) and 82.8% (65.8%-100.0%), respectively (Fig. 3A). Cox proportional hazard analyses did not reveal any independent risk factors for PFS, except for PTI, which was significantly associated with PFS in the univariate proportional hazard analyses (Supplement Table 2). The receiver operating characteristic (ROC) curve showed that the best cutoff value of PTI for predicting PFS was 10. Subgroup analyses revealed that patients with PTI scores >10 had worse PFS (HR = 18.26; 95% CI: 2.60–128.04; $p < 0.001$) than those with

## Table 1 | Baseline Characteristics

| Variables | Total (*N* = 45) |
|---|---|
| Sex, No. (%) | |
| Male | 22 (48.9) |
| Female | 23 (51.1) |
| Age, No. (%) | |
| ≤65 years old | 38 (84.4) |
| >65 years old | 7 (15.6) |
| Charlson Comorbidity Index, No. (%) | |
| ≤3 | 38 (84.4) |
| >3 | 7 (15.6) |
| Pre-treatment MG, No. (%) | |
| Yes | 12 (26.7) |
| No | 33 (73.3) |
| MGFA grade | |
| I | 6 (13.3) |
| II | 4 (8.9) |
| III | 1 (2.2) |
| IV | 1 (2.2) |
| Mode of pleural metastasis, No. (%) | |
| DNT | 12 (26.7) |
| TPR | 33 (73.3) |
| Pathology, No. (%) | |
| A | 1 (2.2) |
| AB | 1 (2.2) |
| B1 | 3 (6.6) |
| B2 | 14 (31.1) |
| B2/B3 mix | 10 (22.2) |
| B3 | 10 (22.2) |
| Thymic carcinoma | 6 (13.3) |
| Pathological stage T, No. (%) | |
| T0 | 31 (68.9) |
| T+ | 14 (31.1) |
| Pathological stage N, No. (%) | |
| N0 | 39 (86.7) |
| N+ | 6 (13.3) |
| Pathological TNM stage, No. (%)# | |
| IVA | 40 (88.9) |
| IVB | 5 (11.1) |
| Tumor location, No. (%) | |
| Left thorax | 11 (24.4) |
| Right thorax | 34 (75.6) |

*CCI* Charlson comorbidity index, *MG* myasthenia gravis, *MGFA* Myasthenia Gravis Foundation of America, *DNT* de novo Masaoka stage IVA TETs, *TPR* TETs with pleural recurrence, *pT+* postoperative pathological T stage positive, *pN+* postoperative pathological N stage positive, *TNM* tumor-node-metastasis.
# Thymic malignancies were graded by AJCC/UICC/IASLC/ITMIG TNM stage (9th edition).

## Table 2 | Surgical outcomes

| Variables | Total (*N* = 45) |
|---|---|
| Surgery, No. (%) | |
| P/D | 3 (6.7) |
| eP/D | 42 (93.3) |
| Resection, No. (%) | |
| R0 | 23 (51.1) |
| R1 | 8 (17.8) |
| R2 | 14 (31.1) |
| Tumor in different zones, No. (%) | |
| I | 41 (91.1) |
| II | 34 (75.6) |
| III | 23 (51.1) |
| IV | 32 (71.1) |
| V | 19 (42.2) |
| PTI, median score [IQR] | 6 [5,9] |
| Combined resection, No. (%) | |
| Diaphragm | 37 (82.2) |
| Lung | 35 (77.8) |
| Pericardium | 7 (15.6) |
| Azygos vein | 1 (2.2) |

*P/D* pleurectomy/decortication, *eP/D* extended pleurectomy/decortication, *R0* complete cytoreductive surgery without residual visible disease, *R1* optimal cytoreductive surgery with residual tumors measuring no more than 10 mm, *R2* incomplete cytoreductive surgery with residual lesions measuring >10 mm in diameter, *PTI* pleural tumor index, *LOS* length of post-operative stay, *IQR* interquartile range.

PTI scores ≤10 (Fig. 3C). In addition, compared with those who underwent R2 resection, patients who underwent R0/1 resection had better PFS; however, this difference was not significant (HR = 0.229; 95% CI: 0.05–1.14; log-rank, $p = 0.130$) (Fig. 3D). Moreover, patients with TPR > 5 years after thymectomy had better PFS than those with TPR ≤ 5 years after thymectomy or those with DNT [2-year PFS: 83.3% (58.3–100.0%) vs. 70.3% (39.4–100.0%) vs. 100.0% (100.0–100.0%)]; however, this difference between two groups was not significant (Fig. 3E). There was no significant difference in PFS (HR = 1.22; 95% CI: 0.12–12.21; log-rank, $p = 0.850$) between patients with pathological stage IVA disease and those with pathological stage IVB disease (Fig. 3F). In addition, no significant difference was observed in PFS between the thymoma and thymic carcinoma patients (Supplement Fig. 9A).

Two patients (4.4%) died before the latest visit; one patient died due to tumor progression in the 34th month after treatment, and the other died due to infectious disease in the 35th month after treatment. The 1- and 2-year OS rates were both 100.0% (100.0–100.0%) (Fig. 3B). There was no significant difference in OS between the thymoma and thymic carcinoma patients (Supplement Fig. 9B). Patients without preoperative treatments did not have significant survival differences compared to those who underwent preoperative treatments (Supplement Fig. 10).

## Discussion

To date, S-HITOC application for TETs with pleural spread or recurrence is mostly based on the experience of clinicians; therefore, the regimens are heterogeneous. Previous reports were retrospective with small sample size, which are not conducive to the formulation of the standard treatment criteria. Therefore, we conducted this phase II clinical trial to evaluate the safety and efficacy of S-HITOC in TETs patients with pleural spread or recurrence. To our knowledge, this is the first study to assess the S-HITOC with uniform regimen in TETs with pleural dissemination or recurrence. Our work revealed that there were manageable treatment related adverse events and no increased need for reoperation or risk of postoperative mortality was recognized. The posttreatment hospitalization duration was accepted. Most patients did not experience intolerable pain after treatment and recovered their quality of life two months after treatment. The latest follow-up was encouraging, with recurrence in 5/45 patients (11.1%) and death in 2/45 patients (4.4%). The 2-year PFS and OS rates were 82.8% and 100.0%, respectively. Therefore, our study revealed that S-HITOC is safe and effective for TETs with pleural spread or recurrence.

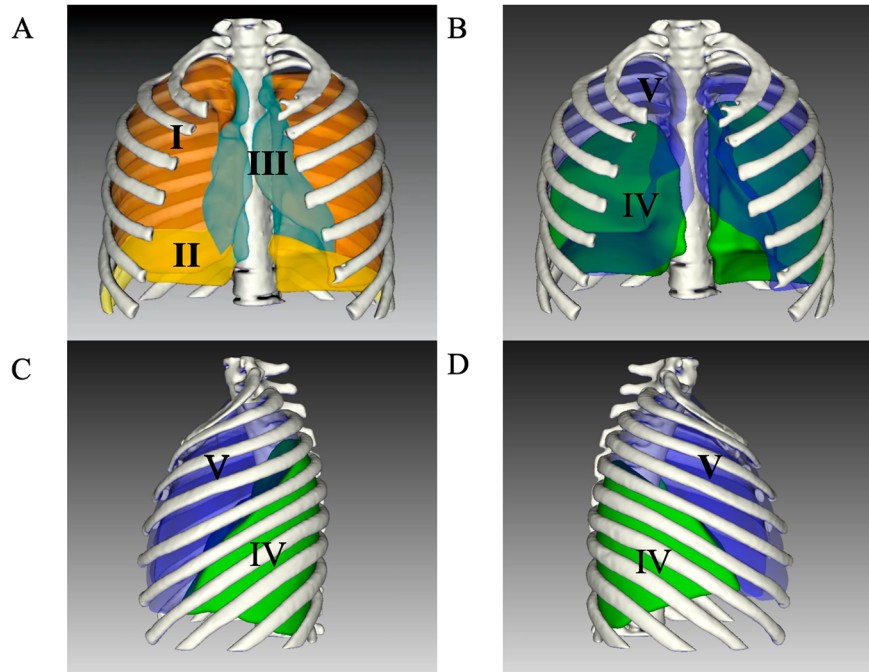

**Fig. 1 | The zone system of pleura for assessment pleural tumor index.**
**A**, **B** Coronal view of the thorax; **C** axial view of the thorax (left); **D** axial view of the thorax (right). Zone I, rib and chest wall zone; Zone II, diaphragm zone; Zone III, mediastinum zone; Zone IV, lower pulmonary zone (below oblique fissure); Zone V, upper pulmonary zone (above oblique fissure).

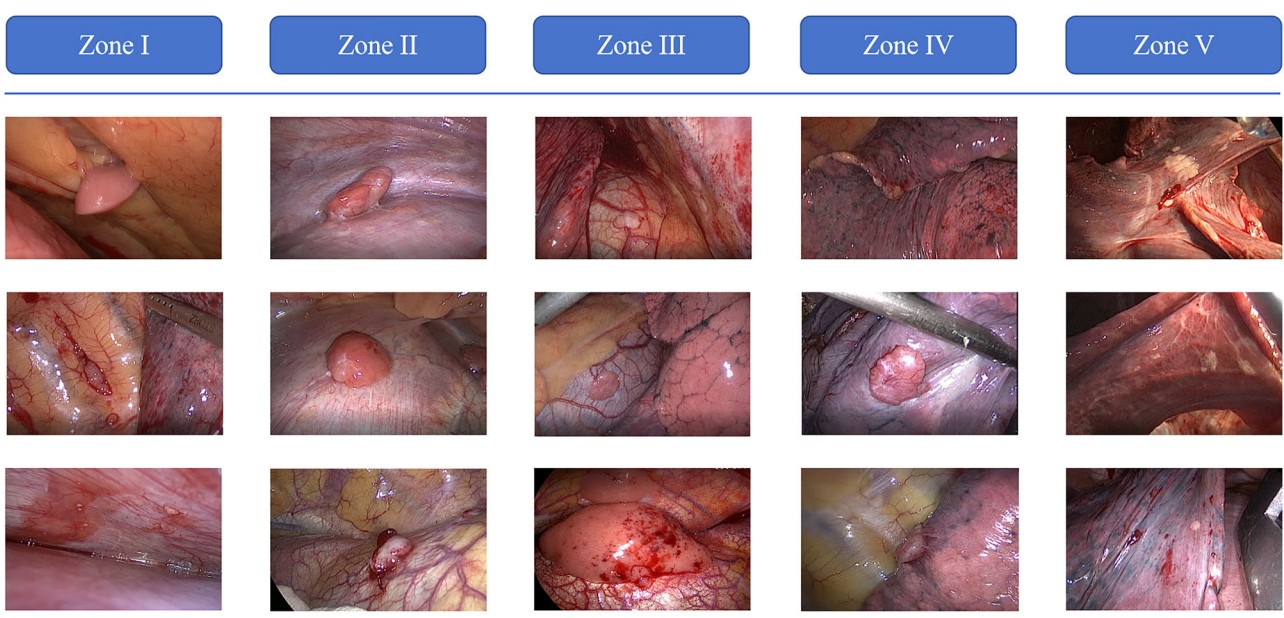

**Fig. 2 | The typical metastatic nodules observed in surgery according to the zone system of pleura.** Zone I, rib and chest wall zone; Zone II, diaphragm zone; Zone III, mediastinum zone; Zone IV, lower pulmonary zone (below oblique fissure); Zone V, upper pulmonary zone (above oblique fissure).

Surgical resection has been recommended as the principal treatment, and completeness of resection is considered the most important determinant of long-term survival in patients with TETs. The present data demonstrated that the prognosis of TET patients with pleural dissemination is not as dismal as expected. In this trial, we removed all macroscopic tumor nodules to the greatest extent; however, there is no consensus on whether a high number of metastatic nodules is a contraindication for cytoreductive surgery. Yano and colleagues reported that a small number of recurrent thymoma lesions were associated with a better prognosis, which indicated that the number of disseminated nodules may be a prognostic factor in Masaoka IVA disease[11]. Additionally, Okuda et al. suggested that the operation would be continued and the nodules would be resected if there were 10 or fewer disseminated nodules in a retrospective study of 136 patients with stage IVA thymoma patients[12]. To date, there is no recognized evaluation system to assess the pleural tumor burden for TETs with

**Table 3 | Treatment-related adverse events**

| Category | Grade 1, No. (%) | Grade 2, No. (%) | Grade 3a, No. (%) | Grade 3b, No. (%) | Grade 4, No. (%) | Grade 5, No. (%) | Any grade, No. (%) |
|---|---|---|---|---|---|---|---|
| Total | 33 (73.3) | 14 (31.1) | 8 (17.8) | 0 (0) | 0 (0) | 0 (0) | 36 (80.0) |
| Anemia | 10 (22.2) | 0 (0) | 0 (0) | 0 (0) | 0 (0) | 0 (0) | 10 (22.2) |
| Fever | 5 (11.1) | 4 (8.9) | 0 (0) | 0 (0) | 0 (0) | 0 (0) | 9 (20.0) |
| Nausea | 4 (8.9) | 0 (0) | 0 (0) | 0 (0) | 0 (0) | 0 (0) | 4 (8.9) |
| Vomiting | 2 (4.4) | 0 (0) | 0 (0) | 0 (0) | 0 (0) | 0 (0) | 2 (4.4) |
| Diarrhea | 2 (4.4) | 0 (0) | 0 (0) | 0 (0) | 0 (0) | 0 (0) | 2 (4.4) |
| Constipation | 7 (15.6) | 0 (0) | 0 (0) | 0 (0) | 0 (0) | 0 (0) | 7 (15.6) |
| AST/ALT increased | 9 (20.0) | 2 (4.4) | 0 (0) | 0 (0) | 0 (0) | 0 (0) | 11 (24.4) |
| Hepatic function abnormal | 2 (4.4) | 0 (0) | 0 (0) | 0 (0) | 0 (0) | 0 (0) | 2 (4.4) |
| Hypoalbuminemia | 6 (13.3) | 0 (0) | 0 (0) | 0 (0) | 0 (0) | 0 (0) | 6 (13.3) |
| Serum creatinine increased | 3 (6.7) | 1 (2.2) | 0 (0) | 0 (0) | 0 (0) | 0 (0) | 4 (8.9) |
| Renal function abnormal | 1 (2.2) | 0 (0) | 0 (0) | 0 (0) | 0 (0) | 0 (0) | 1 (2.2) |
| Pneumonia | 4 (8.9) | 3 (6.7) | 1 (2.2) | 0 (0) | 0 (0) | 0 (0) | 8 (17.8) |
| Pleural effusion | 7 (15.6) | 4 (8.9) | 2 (4.4) | 0 (0) | 0 (0) | 0 (0) | 13 (28.2) |
| Pneumothorax | 4 (8.9) | 3 (6.7) | 2 (4.4) | 0 (0) | 0 (0) | 0 (0) | 9 (20.0) |
| Hemothorax | 0 (0) | 0 (0) | 1 (2.2) | 0 (0) | 0 (0) | 0 (0) | 1 (2.2) |
| Subcutaneous emphysema | 4 (8.9) | 1 (2.2) | 0 (0) | 0 (0) | 0 (0) | 0 (0) | 5 (11.1) |
| Electrocardiogram QT prolonged | 4 (8.9) | 1 (2.2) | 0 (0) | 0 (0) | 0 (0) | 0 (0) | 5 (11.1) |
| Atrial fibrillation | 2 (4.4) | 1 (2.2) | 1 (2.2) | 0 (0) | 0 (0) | 0 (0) | 4 (8.9) |
| Intolerable pain | 0 (0) | 0 (0) | 1 (2.2) | 0 (0) | 0 (0) | 0 (0) | 1 (2.2) |

pleural dissemination, which limits the exploration and development of standardized treatment for TETs with pleural dissemination to a certain extent. Encouraged by the positive role of the peritoneal carcinoma index scoring system for peritoneal carcinoma, we propose the PTI scoring system as a reproducible index for evaluating pleural tumor burden and distribution. In addition, ROC curve analysis showed that PTI scores >10 were associated with worse PFS. However, the results require further verification in a long follow-up period.

Some authors have reported that extrapleural pneumonectomy (EPP) is effective for complete resection of lesions formed via pleural dissemination, especially in patients with pulmonary parenchymal infiltration[3,13]. However, a Japanese national study revealed that 8 of 136 patients with IVA thymoma with pleural spread underwent EPP and the 5-year OS was only 70%[12]. Other studies had reported a higher risk of mortality and morbidity after EPP, ranging from 0% to 17% and 20% to 47%, respectively[3,13,14]. Therefore, the use of EPP should be carefully considered even in "superselected" patients by well-trained teams because of the high operative mortality and low postoperative quality of life. In contrast, P/D and eP/D become increasingly preferred surgical options for patients with pleural tumors that are considered to be lung-preserving and have a high success rate. P/D and eP/D were found to be associated with improved quality of life compared to EPP[15]. P/D and extended resection have also been found to be safe procedures for elderly patients (≥70 years old) without increased morbidity, mortality or short-term survival[16]. Therefore, for patients with TETs with DNT or TPR, cytoreductive surgery may preferably be performed via lung-preserving procedures. In this trial, the complete resection rate was 51.1%, which is not inferior to that reported in previous studies[17,18]. Combination with HITOC could further improve disease control, especially in patients with residual lesions.

The objective of additional HITOC after surgical cytoreduction is to eradicate residual tumor cells to improve local tumor control and patient survival[19]. According to data analyses performed by Aprile et al. and Ambrogi et al., surgery without HITOC is often associated with higher recurrence rates and shorter OS than surgery with HITOC[10,20]. Various chemotherapeutic regimens for TETs have been reported; however, there is no standard chemotherapeutic protocol for HITOC.

Cisplatin is the most common component of the perfusion drug. However, the doses of cisplatin varied. A previous study showed that 50 mg cisplatin for 60 min achieved a functional maximum penetration depth after P/D[21]. Notably, the application of higher doses of cisplatin seems to be associated with better OS, but might also cause more adverse events, especially renal dysfunction, which occurred in 3.3–57% of patients in previous studies[22–24]. Additionally, a two-drug regimen of HITOC is preferred because of its synergistic antitumor effects, and doxorubicin at a dose ranging from 15 to 100 mg/m² is often recommended, as one of the classic chemotherapy drugs for advanced thymic tumors[25]. Furthermore, the antitumor effect of chemotherapeutic drugs is enhanced when administered at elevated temperatures[26]. In addition, hyperthermia alone has a direct cytotoxic effect on malignant cells; at temperatures greater than 40 °C, hyperthermia stimulates apoptosis, inhibits angiogenesis, impairs DNA repair, denatures proteins, and upregulates heat shock proteins that stimulate natural killer cells[27]. However, maintaining a temperature of ≤43 °C is crucial because of the risk of pulmonary edema. Therefore, we conducted the HITOC protocol, which involved treatment with 50 mg/m² cisplatin and 25 mg/m² doxorubicin for 60 min at 43 °C. Our study revealed that, under our HITOC protocol, no patient developed severe cisplatin-related toxicity without nephroprotection. The promising 2-year PFS rate of 82.8% and OS rate of 100.0% among our patients further confirm the feasibility and efficacy of our protocol.

In our study, the rates of grade ≥3 treatment related adverse events and mortality were 17.8% and 0%, respectively. This compares favorably with figures reported in other series, with mortality rates ranging from 0% to 17%[28], and is mostly related to the extent of resection and the use of combined chemotherapeutic regimens with low-dose drug combinations. The overall treatment related adverse events were similar to those reported in a recent systematic literature review[25]. Most patients in our cohort received eP/D; however, treatment-related adverse events were accepted and not associated with delayed HITOC or prolonged hospital stay. Notably, pain improvement and QoL recovery, which have not been reported in previous studies, were observed two months after treatment under our protocol. In addition, we identified a potential function of our

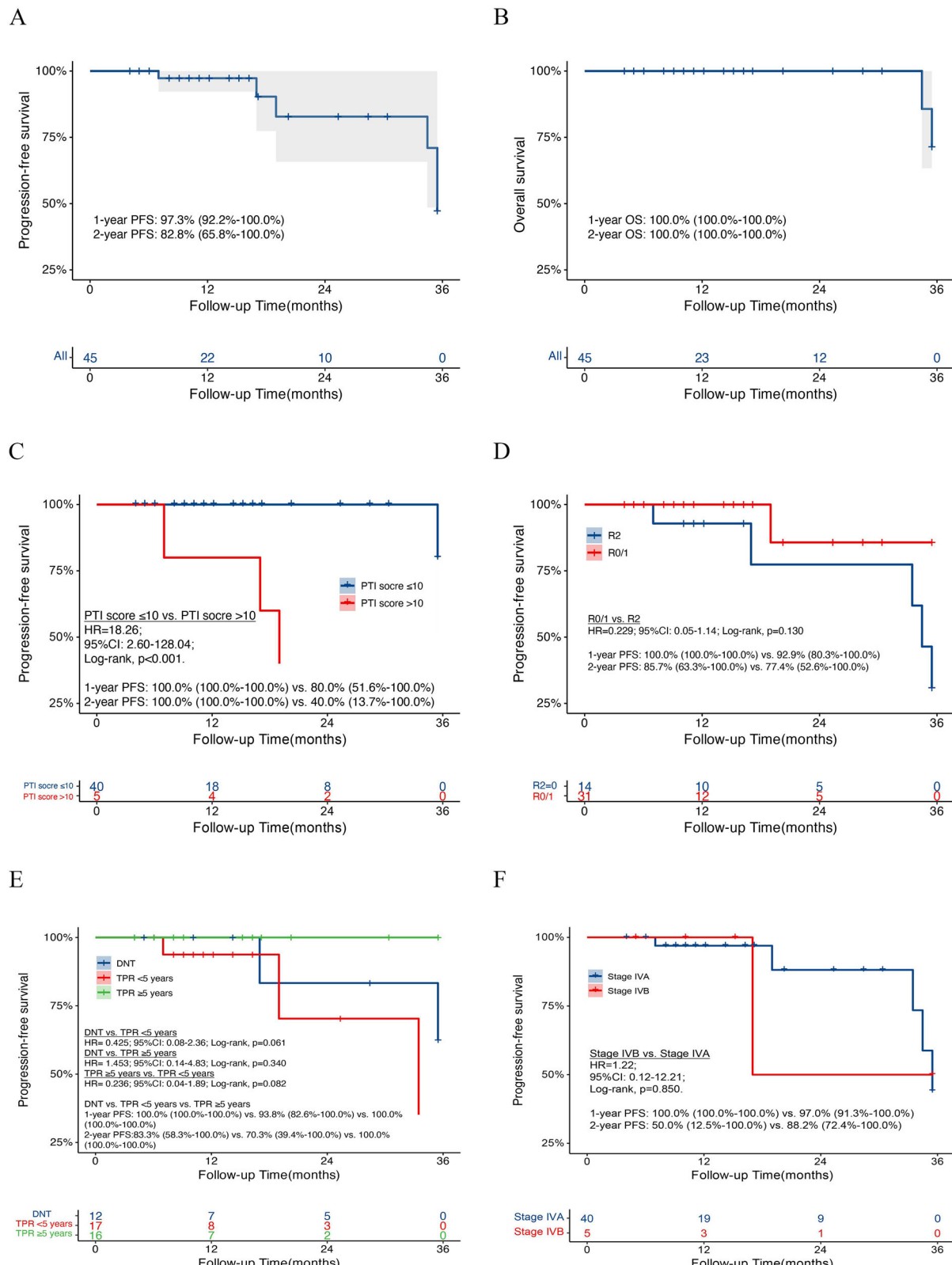

**Fig. 3 | Survival analysis of patients with S-HITOC. A** PFS among all patients (*N* = 45); **B** OS among all patients (*N* = 45); **C** PFS based on the PTI score [*N* = 45, N (PTI score > 10) = 5, *N* (PTI score ≤ 10) = 40]; **D** PFS based on the surgical resection completeness [*N* = 45, *N* (R2) = 14, *N* (R0/R1) = 31]; **E** PFS based on the pleural dissemination classification [*N* = 45, *N* (DNT) = 12, *N* (TPR < 5 year)=17, *N* (TPR ≥ 5 year) =16]; **F** PFS based on the tumor stage [*N* = 45, *N* (IVA) = 40, *N* (IVB) = 5]. PFS progression-free survival, OS overall survival, PTI pleural tumor index, R0 complete cytoreductive surgery without residual visible disease, R1 optimal cytoreductive surgery with residual tumors measuring no more than 10 mm, R2 incomplete cytoreductive surgery with residual lesions measuring > 10 mm in diameter, DNT de novo Masaoka stage IVA TETs, TPR TETs with pleural recurrence, HR hazard ratio, CI confidence interval.

S-HITOC regimen in the improvement of MG, with CSR achieved in 25.0% of the cases. Notably, all these MG patients achieved reduced glucocorticoid dose after S-HITOC. However, further confirmation through randomized controlled trials is required.

Our findings must be considered within the context of several limitations. First, as a single-arm, open, single-institution trial, it reflects only the experience of a single specialized institution. The patients in this study were all Chinese Han population, which has limited ethnic diversity. Therefore, our findings may not be generalizable to other settings and ethnic groups. Second, the sample size was relatively small because of the rarity of TETs. Also, the fact that some patients received preoperative treatment might exaggerate the survival benefit of S-HITOC for TETs. The subgroup analysis showed that patients who did not receive preoperative treatment had similar PFS to those who received preoperative treatment (Supplement Fig. 10). However, survival bias from different preoperative treatments could not be neglected in this primary trial. The lack of a control group precludes any comparative analysis to isolate the effects of surgical intervention or HITOC, and a randomized controlled trial is further required. PTI was used to describe pleural tumor burden and screen beneficiaries of S-HITOC. However, it was not suitable for assessment in other issues (such as pericardial dissemination). The incidence of pericardial dissemination is even rarer, and further patient accumulation is needed to explore the characteristics and treatment options for patients with pericardial dissemination. Also, patients with PTI scores >10 had worse PFS than those with PTI scores ≤10. Patients within this subgroup represent a distinct and challenging cohort within the broader population of TETs. Current treatment modalities yield limited clinical benefits for this subset, and there is a pressing need to explore additional therapeutic approaches for those with PTI scores >10. Finally, this study just reported short-term outcomes with the limited follow-up time. It is insufficient to draw a very reliable conclusion. Long-term follow-up is needed to confirm the safety (cardiac function, etc.) and benefits of S-HITOC. Further validation of our conclusions through multi-center, large-scale prospective trials is essential.

In summary, this study offers evidence of S-HITOC's efficacy and safety for TETs with pleural spread or recurrence. S-HITOC emerges as a viable therapeutic candidate for TETs with pleural spread or recurrence.

## Methods

### Ethics statement
This research study complied with all relevant ethical regulations. The trial was conducted in accordance with the criteria set by the Declaration of Helsinki. The trial was approved by the Zhongshan Hospital Research Ethics Committee (ID: B2021-703R). This trial was registered at ClinicalTrials.gov (identifier: NCT05446935). All the investigators provided informed consent from each participant. This study was an open, single-arm, phase II trial that aimed to evaluate the perioperative safety and efficacy of S-HITOC in the treatment of TETs with pleural spread or recurrence (Supplement Note 2). The trial protocol has been published[29].

### Patients
Between August 1, 2021, and February 29, 2024, 45 patients who received S-HITOC at Zhongshan Hospital of Fudan University were included in this work. We confirm that the registered trial is fully closed. Although we would further follow up with the patients and take responsibility for postoperative care, we confirm all secondary outcomes are reported in this work.

Eligible patients were adults with TETs confirmed by pathological examination and pleural spread or recurrence diagnosed by imaging, regardless of ethnicity. Patients were excluded if they had an acute exacerbation of myasthenia gravis, renal dysfunction, a performance status score of more than 2, other malignant carcinomas, allergy to cisplatin or doxorubicin, or refused to participate in the study.

### Reporting on sex and gender
In this study, we utilized self-reported sex as a basis for our analysis. The selection of participants was made without regard to sex or gender, with the primary aim of maximizing the sample size. It is important to note that sex was not analyzed in separate groups, nor was it taken into consideration during the statistical testing process. Within the cohort, 51.1% of the participants identified themselves as female.

### Treatments
Surgery was performed on all enrolled participants to reduce the tumor burden. The most appropriate surgical method was selected based on the location and number of lesions, and all visible tumor lesions were removed as completely as possible. P/D has been used to treat oligometastatic pleural nodules. eP/D was defined as P/D with additional partial resection of the lungs, diaphragm, pericardium, and/or azygos vein. HITOC was performed using a BR-TRG-I-type device [Guangzhou Bright Medical Technology, Guangzhou, China], with a circulation flow of 400 mL/min and an inflow temperature of 43 °C. Doxorubicin was administered at a dose of 25 mg/m² on postoperative day 1. Cisplatin [50 mg/m² on postoperative day 2 (posttreatment day 0)]. The inflow chest tube was not removed, and excess pleural fluid was allowed to flow freely through the chest tube to the collecting system after HITOC was performed for 60 min.

### Pleural tumor index
The pleural tumor burden was assessed by dividing the pleural cavity into five zones during surgery: Zone I, the rib and chest wall zone; Zone II, the diaphragm zone; Zone III, the mediastinum zone; Zone IV, the lower pulmonary zone (below the oblique fissure); and Zone V, the upper pulmonary zone (above the oblique fissure). The largest lesion in each zone was scored from 0 to 3, ranging from no gross lesion to extensive lesions (grade 0, no visible lesions; grade 1, lesion diameter <1.0 cm; grade 2, lesion diameter between 1.0–5.0 cm; and grade 3, lesion diameter >5.0 cm or lesion fusion). The final score is the sum of the scores for each zone, ranging from 0 to 15 points: 0 points for no visible tumor lesions and 15 points for the highest tumor burden. Resection status was assessed by resection extent of visible pleural nodules, namely: R0, complete cytoreductive surgery without residual visible disease; R1, optimal cytoreductive surgery with residual tumors measuring no more than 10 mm; R2, incomplete cytoreductive surgery with residual lesions measuring >10 mm in diameter.

### Postoperative care and follow-up
All adverse events and postoperative complications were recorded and treated. VAS scores were recorded to evaluate pain by making a handwritten mark on a 10-cm line (VAS ruler) before the operation and at 1 and 3 days after treatment. The QoL was evaluated at baseline and after treatment on the 1st, 30th and 60th days by using the European Organization for Research and Treatment of Cancer Quality of Life Questionnaire C-30 Scale (EORTC QLQ-C30) (V3.0). MG was classified according to the MGFA clinical classification system. The prognosis was evaluated according to the MGFA post-intervention status, and overall remission included CSR, PR, and MM. For all patients who received S-HITOC, chest computed tomography scans were performed every 3 months for the first 6 months after treatment, then every 6 months for the first two years and finally annually throughout their lifetime. Further examinations, including ultrasound, puncture biopsy, and positron emission tomography-computed tomography, were performed when needed.

### Endpoints and assessments

The primary endpoint was the major treatment related adverse events, defined as grade ≥3, according to the Clavien–Dindo classification (5th edition) and the Common Terminology Criteria for Adverse Events, Version 5.0 (CTCAE v5.0). The secondary outcomes included the LOS, EORTC QLQ-C30 QoL score, VAS score, PFS, and OS.

### Statistics and reproducibility

The sample size was based on estimates of major treatment related adverse event rates in patients who underwent S-HITOC. A major treatment related adverse events rate within 15% was considered manageable and a rate greater than 30% was considered unsafe. Therefore, 37 patients were required at a $p = 0.1$ significance level with 80% power to detect a 15% difference in the major treatment related adverse events rate. The experiments were not randomized. The investigators were not blinded to allocation during experiments and outcome assessment.

The beginning of follow-up was defined as the date of cytoreductive surgery regardless of whether preoperative treatment was used. The Mann-Whitney test was used to compare the EORTC QLQ-C30 QoL scores and VAS scores at baseline and after treatment, and the Student's t-test was used to compare glucocorticoid usages. Progression was defined as documented intrathoracic and/or extrathoracic, ipsilateral and/or contralateral tumor detection by cytology/histology and/or imaging. PFS was defined as the time from surgery to the first objective tumor progression or death from any cause, whichever occurred first. OS was defined as the time from surgery to death due to any cause. The cut-off date for the last follow-up was June 30, 2024. PFS and OS were calculated using the Kaplan-Meier method with the log-rank test. The Cox proportional hazards regression model was used to explore the potential risk factors of PFS, and variables with $p$ values < 0.25 in the univariate analysis were entered into the multi-variable analysis. The ROC curve was used to calculate the best cutoff value of the PTI for predicting PFS. The R software (Version 4.1.2, The R Foundation, Vienna, Austria) and GraphPad Prism software (Version 8.0, GraphPad Software, Inc., CA) were used for data analysis and image creation.

### Reporting summary

Further information on research design is available in the Nature Portfolio Reporting Summary linked to this article.

## Data availability

Source data are provided as a Source Data file without identifying patient's individual information. The data generated in this study have been deposited in the Figshare repository database without accession code. They can be freely and enduringly accessed (Figshare private link: https://figshare.com/s/c146c8764027700ee69a). Source data are provided with this paper.

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

## Acknowledgements

We acknowledge the trust and partnership of the anesthesiology members and study participants in engaging with us in this trial. The authors gratefully acknowledge funding by National Key Research and Development Program of China (No. 2023YFC3402700), awarded to J.D. and funding by National Natural Science Foundation of China, awarded to J.D.(No.82472924) and S.W. (No. 82404059). The study was supported by National Key Research and Development Program of China (No. 2023YFC3402700), awarded to J.D. and funding by National Natural Science Foundation of China, awarded to J.D. (No.82472924) and S.W. (No. 82404059). ClinicalTrials.gov identifier: NCT05446935.

## Author contributions

J.D., S.W. and L.T. conceived and designed the protocol for this study. X.Y., J.J., F.L., Y.Z., Y.A. and J.G. collected the clinical data. H.W. and S.W. supervised and validated the data and administrated the project. F.L. and X.Y. curated and analyzed the data. J.J. and Y.A. contributed to data acquisition and exploratory analyses. F.L., Y.A. and J.G. interpreted the data. S.W. and X.Y. wrote the first draft, and J.J. and H.W. reviewed and edited the manuscript. All authors contributed to reviewing and revising the manuscript and approved the final version.

## Competing interests

The authors declare no competing interest.
