## [Transparent Peer Review file · Nature Communications]

Cytoreductive Surgery and Hyperthermic Intrathoracic Chemotherapy in Thymic Epithelial Tumors with Pleural Spread or Recurrence: A Prospective, Single-Arm, Phase II Study

Corresponding Author: Professor Jianyong Ding

Version 1:

Reviewer comments:

Reviewer #1

(Remarks to the Author)

Hyperthermic intrathoracic chemotherapy (HITHOC) is a treatment option for thymic epithelial tumors because the high temperature and concentration of cytostatic drugs can induce tumor cell apoptosis and activate the immune system. Some authors have reported that cytoreductive surgery and HITOC (S-HITOC) is effective on survival for patients with pleural metastatic thymic tumors. In this paper, the authors presented the results of a prospective, single-arm, phase II study, in which treatment regimens were uniformly decided and 45 patients were enrolled. Whereas I think this paper is an important contribution, there are some concerns as shown below.

1. Patient characteristics (Table 1): The WHO type of thymomas should be shown. In general, surgery for stage IVa thymic cancer is contraindicated, thus perioperative treatment or treatment after recurrence should be shown and discussed in detail.
2. Whereas pleural tumor index (PTI) seems to be useful for evaluating the pleural tumor burden, how can we evaluate the number of disseminated nodules? In addition, how can we indicate pericardial dissemination using PTI?
3. Surgical treatment: All patients underwent P/D or eP/D? How about partial pleurectomy which is often performed for thymoma patients with localized pleural dissemination. Were the patients with massive pleural disseminations enrolled into the study? In addition, for patients with de novo Masaoka stage IVa diseases, how did the authors perform surgery (surgical approach etc.)? Was it possible for patients with air leak to perform HITHOC?
4. Is HITHOC feasible after pericardial resection? I think longer follow-up (cardiac function etc) should be performed.
5. The follow-up period is also insufficient to evaluate the treatment effectiveness in thymoma because of its slow progression. I think patients with R2 resection should be excluded for evaluation of progression free survival (PFS).
6. The S-HITOC treatment for thymic tumor is performed for patients with massive pleural disseminations. However in this paper, patients with high PTI scores had worse PFS. What kind of patients with thymic epithelial tumors can we use this procedure for?
7. For patients with thymoma, recurrence does always not lead to death and deaths are often not due to recurrence.

Reviewer #2

(Remarks to the Author)

Dear authors, thank you for giving me the opportunity to review your article. The study investigated cytoreductive surgery combined with hyperthermic intrathoracic chemotherapy in treating stage IV thymic epithelial tumors with pleural spread or recurrence. You reported a 2-year progression-free survival rate of 82.8% and a 100% overall survival rate. The treatment demonstrated acceptable levels of complications, manageable pain and post-operative quality of life, and an important remission rate for patients with myasthenia gravis. You developed the Pleural Tumor Index which was correlated with PFS. This is the first prospective, single-arm phase II study that could potentially establish S-HITOC as a future standard care option.

Some comments:

- I should include the WHO classification of the thymoma and correlate with survival.
- Why did you include thymic carcinoma on your study? This is clearly not the same disease than thymoma.
- Generally speaking, thymoma is an indolent tumour with slow growing recurrence. In my opinion, OS or even PFS are difficult to interpret for this particular tumour. You should need to publish late results. For me, it is difficult to make any conclusion without comparative groups, because after 1 year (your median follow-up), I would be sure that the results would be similar with or without HITOC.. This point should be explained on the title (initial peri-operative outcomes or morbidity of such procedure)
- I like the idea of PTI score's prognostic value. This is a good idea to better compare the patients that are quite heterogeneous
- Sufficient detail is provided for reproducibility, including precise dosages, chemotherapy temperatures, and surgical techniques.

Reviewer #3

(Remarks to the Author)

Firstly, I would like to congratulate the authors on their efforts to study a relatively large series of patients with this rare condition.

Secondly, this study does add to the existing literature on this subject.

I have the following comments which I would ask the authors to address -

1. As this was a prospective study, in the same time frame how many were assessed for this treatment but were not found to be suitable and why was that?
2. I note the high number of patients (48.9%) who had residual tumour left after surgery. Was the tumour thought to be fully resectable in these patients preoperatively or was the intent just debulking?
3. Can you describe the surgical technique in more detail? Why was tumour left in nearly half of the patients? Was there an intentional limit to your extent of surgery ? pneumonectomy
4. Can you clarify your definition of R1 as <1cm residual tumour? Where is this definition from?
5. Can the authors be more specific in the future selection of patients for this invasive technique? Are there clear exclusion criteria from their data and experience?

Reviewer #4

(Remarks to the Author)

This trial evaluated outcomes of cytoreductive surgery with hyperthermic intrathoracic chemotherapy (S-HITOC) for thymic epithelial tumors (TETs) with pleural spread or recurrence. Between August 2021 and February 2024, 45 patients underwent surgery followed by HITOC with doxorubicin and cisplatin. Complete resection was achieved in 51.1% of cases, with a Clavien-Dindo grade ≥ 3 complication rate of 17.8%. At a median follow-up of 12.2 months, 2-year progression-free survival (PFS) and overall survival (OS) were 82.8% and 100%, respectively. Patients with a pleural tumor index (PTI) >10 had significantly worse PFS. Although HITOC demonstrated promising oncological benefits and effective myasthenia gravis symptom control, several issues remain regarding study design and conclusions:

1. As a single-center, small-sized study, caution is necessary when interpreting these findings. The sample size is justified by a reduction in treatment-related adverse events (AEs) from 30% to 15%, but several questions arise:
 - o How are surgery-related AEs defined? Is this specific to certain AEs or pooled together?
 - o No formal test against the 30% threshold is presented, raising concerns about the relevance of this sample size justification for the primary analysis.
 - o No rationale or benchmark is provided for selecting the 30% and 15% thresholds.
 2. In a single-arm study, benchmark values are essential to contextualize outcomes. Clear definitions of favorable surgical outcomes, acceptable AE rates, and survival expectations are lacking in the manuscript.
 3. Certain analyses appear missing from the "Statistical Analysis" section, such as details on how p-values in supplementary figures 6, 7, and 9 were obtained.
 4. Some claims, such as the absence of selection bias (line 294), may be overstated. Given that this is a single-center, single-arm study, selection bias is possible.
 5. The manuscript does not adequately discuss the generalizability of findings to other ethnic groups.
 6. Without a concurrent control group, it is challenging to draw conclusions about the specific effects of the surgical intervention or HITOC. More justification is needed to support the efficacy claims made in the study.
- These points highlight areas where the manuscript could be strengthened for clearer interpretation and reliability.

Version 2:

Reviewer comments:

Reviewer #1

(Remarks to the Author)

The authors presented the prospective, single-arm, phase II study to investigate the safety and short-term survival of cytoreductive surgery followed by hyperthermic intrathoracic chemotherapy (S-HITOC) for 45 TETs patients with

pleural spread or recurrence. They concluded that early clinical outcomes were good, and S-HITOC emerges as a viable therapeutic candidate for TETs with pleural spread or recurrence. This second version of the paper is a great improvement. Pleural dissemination is a common pattern of failure after initial treatment of thymoma and thymic carcinoma, but there is no standardized treatment, thus I think this paper is an important contribution.

Reviewer #2

(Remarks to the Author)

Dear authors,
thank you for your responses. You addressed all points and suggestions.

Reviewer #4

(Remarks to the Author)

Reviewer #5

(Remarks to the Author)

Thank you for the opportunity to review your article. This is the first prospective, single-arm phase II study that may potentially establish S-HITOC as a future standard of care. I have the following concerns for the authors to address:

1. The threshold of 10 for the PTI was derived using ROC, a traditional method for generating cutpoints. However, how should this threshold be interpreted in terms of disease severity? Specifically, could it be used to define 'massive pleural dissemination'? Moreover, only 5 patients had a PTI greater than 10. Would this represent a significant subset within the population of all patients with TETs, especially considering that the patients were from a leading hospital in China?
2. When the primary outcome was set as the safety of S-HITOC, specifically targeting an AE rate of 15%, it might be more appropriate to justify the sample size based on estimating an AE rate of 15% with the upper 95% confidence limit below certain clinically meaningful threshold, rather than aiming to detect a difference between 15% and 30% (considered the unsafe threshold).
3. Please specify whether pain was measured using the Visual Analogue Scale (VAS) or the 0-10 Numerical Rating Scale (NRS). When using the VAS, patients visualize their pain on a 10 cm line presented on paper. If patients rated their pain on a 0-10 scale by choosing one of 11 discrete options, then the measure used was the NRS.
4. Please confirm whether the QoL subscale of the EORTC QLQ-C30 was analyzed as the QoL outcome.
5. Additionally, QoL and pain scores typically do not follow a normal distribution. Therefore, non-parametric methods, such as the rank sum test, should be considered instead of the t-test.

Version 3:

Reviewer comments:

Reviewer #5

(Remarks to the Author)

Thank you, the authors have addressed all my concerns.

Response to Reviewer #1 (Thoracic Surgery, clinical)

Hyperthermic intrathoracic chemotherapy (HITHOC) is a treatment option for thymic epithelial tumors because the high temperature and concentration of cytostatic drugs can induce tumor cell apoptosis and activate the immune system. Some authors have reported that cytoreductive surgery and HITOC (S-HITOC) is effective on survival for patients with pleural metastatic thymic tumors. In this paper, the authors presented the results of a prospective, single-arm, phase II study, in which treatment regimens were uniformly decided and 45 patients were enrolled. Whereas I think this paper is an important contribution, there are some concerns as shown below.

Q1. Patient characteristics (Table 1): The WHO type of thymomas should be shown. In general, surgery for stage IVa thymic cancer is contraindicated, thus perioperative treatment or treatment after recurrence should be shown and discussed in detail.

Answer: Thank you very much for your insightful and valuable comments. In response to your suggestions, we have incorporated the WHO classification of thymoma into **Table 1**. This addition is intended to provide patients with a clearer understanding of the characteristics of our patient cohort, as detailed below: Six patients were thymic carcinoma. Among 39 thymoma patients, 1 (2.2%), 1 (2.2%), 3 (6.6%), 14 (31.1%), 10 (22.2%) and 10 (22.2%) patients were type A, AB, B1, B2, B2/B3 mix, and B3, respectively (Table 1). We believe that this modification will enhance the comprehensibility of our data and contribute to a more thorough understanding of the patient demographics in our study.

Revised in the Manuscript: Page 5, line 125-126.

Revised in the Manuscript: Table 1.

Table 1. Baseline Characteristics

Variables	Total (N=45)
Sex, No. (%)	
Male	22 (48.9)
Female	23 (51.1)
Age, No. (%)	
≤65 years old	38 (84.4)
>65 years old	7 (15.6)
Comorbidity, No. (%)	
Yes	19 (42.2)

	No	26 (57.8)
Charlson Comorbidity Index, No. (%)		
	≤3	38 (84.4)
	>3	7 (15.6)
Pre-treatment MG, No. (%)		
	Yes	12 (26.7)
	No	33 (73.3)
MGFA grade		
	I	6 (13.3)
	II	4 (8.9)
	III	1 (2.2)
	IV	1 (2.2)
Mode of pleural metastasis, No. (%)		
	DNT	12 (26.7)
	TPR	33 (73.3)
Pathology, No. (%)		
Thymoma		39 (86.7)
	A	1 (2.2)
	AB	1 (2.2)
	B1	3 (6.6)
	B2	14 (31.1)
	B2/B3 mix	10 (22.2)
	B3	10 (22.2)
Thymic carcinoma		6 (13.3)
Pathological stage T, No. (%)		
	T0	31 (68.9)
	T+	14 (31.1)
Pathological stage N, No. (%)		
	N0	39 (86.7)
	N+	6 (13.3)
Pathological TNM stage, No. (%)		
	IVA	40 (88.9)
	IVB	5 (11.1)
Tumor location, No. (%)		
	Left thorax	11 (24.4)
	Right thorax	34 (75.6)

To date, the prevailing view holds that stage IVa thymic epithelial tumors (TETs) are contraindicated for surgical intervention. However, both the literature and our clinical experience suggest that TETs with pleural dissemination tend to have a more favorable prognosis compared to other chest stage IVa tumors, such as lung cancer and esophageal cancer. Given the benefits of surgery

in reducing tumor burden, numerous domestic and international surgeons have explored cytoreductive surgery as an integral component of the treatment plan for TETs with pleural dissemination [1-4].

Drawing on our prior experience, we posit that cytoreductive surgery followed by hyperthermic intrathoracic chemotherapy (S-HITOC) may offer good safety and feasibility for TETs with pleural dissemination. Consequently, we designed this trial. All enrolled patients were confirmed to have thymic epithelial tumors via preoperative biopsy and pathological examination.

Among the 12 patients with de novo Masaoka stage IVA TETs, 5 (41.7%) did not undergo any preoperative treatment. One (8.3%), 3 (25.0%), and 3 (25.0%) patients received radiotherapy, chemotherapy, and chemoradiotherapy, respectively. Among the 33 TETs with pleural recurrence, 11 (33.3%), 2 (6.1%), and 10 (30.3%) patients received radiotherapy, chemotherapy, and chemoradiotherapy, respectively, following their initial surgery. As a result, 25 (75.8%) did not receive any post-recurrence treatment prior to S-HITOC. Four (12.1%), 1 (3.0%), and 3 (9.1%) patients received radiotherapy, chemotherapy, or chemoradiotherapy, respectively, after recurrence. The detailed data were showed in **Supplement Table 1**.

Supplement Table 1. History of treatments in the all enrolled patients.

Variables	N (%)
DNT (N=12)	
Preoperative treatment	
None	5 (41.7)
Radiotherapy	1 (8.3)
Chemotherapy	3 (25.0)
Chemoradiotherapy	3 (25.0)
TPR (N=33)	
Adjuvant therapy after first surgery	
None	10 (30.3)
Radiotherapy	11 (33.3)
Chemotherapy	2 (6.1)
Chemoradiotherapy	10 (30.3)
Post-recurrence treatment	
None	25 (75.8)
Radiotherapy	4 (12.1)
Chemotherapy	1 (3.0)
Chemoradiotherapy	3 (9.1)

Strictly speaking, if the role of S-HITOC for TETs with pleural dissemination is to be thoroughly explored, patients who received preoperative treatment should ideally be excluded. However, given the rarity of such cases, we did not consider preoperative treatment as one of the exclusion criteria in this trial. Additionally, we conducted a subgroup analysis of the enrolled patients, and the results revealed that patients who did not receive preoperative treatment had a similar progression-free survival (PFS) compared to those who underwent preoperative treatment (**Supplement Figure 11**). We have also emphasized this limitation in the discussion section to provide a comprehensive context for our findings as followed: Also, the fact that some patients received preoperative treatment might exaggerate the survival benefit of S-HITOC for TETs. The subgroup analysis showed that patients who did not receive preoperative treatment had similar PFS to those who received preoperative treatment (Supplement Figure 11). However, survival bias from different preoperative treatments could not be neglected in this primary trial.

Thank you for your valuable suggestion, which has helped us to further clarify the rationale behind our study design and the interpretation of our results.

Supplement Figure 11. PFS between the patients with preoperative treatment and patients without preoperative treatment.

Revised in the Manuscript: Page 5, line 124-125; Page 8, line 206-207; Page 10-11, line 293-297.

Revised in the Manuscript: Supplement Table 1, Supplement Figure 11.

Reference

1. Marulli G, et al. Surgical treatment of recurrent thymoma: is it worthwhile?†. *Eur J Cardiothorac Surg.* 49,327-332 (2016)
2. Qi W, et al. The role of surgery in advanced thymic tumors: A retrospective cohort study. *Front Oncol.* 12,1073641 (2023)
3. Billè A, et al. Improving outcomes of surgery in advanced infiltrative thymic tumors: the benefits of multidisciplinary approach. *Tumori.* 108,477-485 (2022)
4. Wang S, et al. Induction Therapy Followed by Surgery for Unresectable Thymic Epithelial Tumours. *Front Oncol.* 11,791647 (2022)

Q2. Whereas pleural tumor index (PTI) seems to be useful for evaluating the pleural tumor burden, how can we evaluate the number of disseminated nodules? In addition, how can we indicate pericardial dissemination using PTI?

Answer: Thank you for your sincere inquiry. The Pleural Tumor Index (PTI) was described in detail in the initial manuscript. To assess the pleural tumor burden, the pleural cavity is divided into five zones during surgery: Zone I encompasses the rib and chest wall area; Zone II covers the diaphragm area; Zone III is the mediastinum area; Zone IV includes the lower pulmonary area (below the oblique fissure); and Zone V is the upper pulmonary area (above the oblique fissure). The largest lesion in each zone is scored on a scale from 0 to 3, reflecting the range from no gross lesion to extensive lesions (grade 0 for no visible lesions; grade 1 for lesion diameter <1.0 cm; grade 2 for lesion diameter between 1.0–5.0 cm; and grade 3 for lesion diameter >5.0 cm or lesion fusion). The final score is the sum of the scores for each zone, ranging from 0 to 15 points: 0 points indicate no visible tumor lesions, while 15 points represent the highest tumor burden.

In conceptualizing the PTI, we drew inspiration from the Peritoneal Cancer Index (PCI). Generally, the number of metastatic nodules is a crucial indicator of tumor dissemination. However, several practical issues need to be addressed in real-world studies. The first clinical question pertains to the calculation accuracy of the number of metastatic nodules. In reality, the calculation may lack

precision or objectivity due to the presence of minimally visible nodules. Based on our current experience, preoperative imaging techniques often underestimate the extent of pleural spread, leading us to currently not recommend the use of imaging (CT) for PTI evaluation. In the future, we plan to further investigate whether the combination of multiple imaging methods can enhance the sensitivity and specificity of PTI prediction.

The second clinical question relates to the cutoff number of metastatic nodules between different hierarchies. Establishing consensus-based and widely recognized cutoff number of tumor nodules to distinguish different pleural metastatic classifications is challenging. To date, the international standard value for the number of pleural metastatic lesions to differentiate PTI classifications has not been defined.

The third clinical question concerns whether the concept of the number of pleural metastases could enhance the prognostic discrimination ability of the PTI classification. In the process of evaluating the number of metastatic nodules, it is necessary to measure both the number and size of each metastatic lesion. Surgeons must subjectively assess the number and size of each lesion. Encountering numerous lesions during surgery demands significant labor and time. Moreover, it is uncertain whether such an investment is worthwhile. Previous studies have shown no survival difference according to the varying number of metastatic lesions [1-4]. Classifications using different cutoff numbers of lesions did not effectively improve the ability to discriminate prognosis [1-4].

Taking these considerations into account, we endeavored to design the PTI concept, aiming to enhance the objectivity of the PCI classification while preserving its benefits and simplicity, as well as creating a clear, objective model. We minimized the impact of the number of metastatic nodules. Consequently, in this index, we only evaluate the characteristics of the largest metastatic nodule in each zone and calculate the PTI. This score is straightforward, easily comprehensible, and offers a certain degree of discrimination for pleural dissemination.

It is important to note that PTI is a novel concept we proposed to assess the extent of pleural dissemination in thymic tumor patients. Pleural dissemination is a primary form of thymic tumor metastasis, and this trial focuses on its treatment. The PTI concept was introduced to help us objectively and comprehensively evaluate the extent of pleural dissemination, making it unsuitable for pericardial dissemination. This represents a limitation of the PTI index, which we have acknowledged in the revised manuscript. The incidence of pericardial dissemination of thymic tumors is even rarer,

and further patient accumulation is needed to explore the characteristics and treatment options for patients with pericardial dissemination.

We hope this explanation addresses your concerns. Thank you again for your valuable feedback.

Revised in the Manuscript: Page 3, line 64-65; Page 5, line 136-138; Page 8, line 226-228; Page 8-9, line 233-237; Page 11, line 298-302; Page 12, line 339-349.

Reference:

1. Kawasaki M, et al. Investigation of the Japanese Classification of Peritoneal Metastasis from Colorectal Cancer Referring to the Correlation with PCI. *J Anus Rectum Colon.* 4,157-164 (2022)
2. Shida D, et al. Long-Term Outcomes After R0 Resection of Synchronous Peritoneal Metastasis from Colorectal Cancer Without Cytoreductive Surgery or Hyperthermic Intraperitoneal Chemotherapy. *Ann Surg Oncol.* 25,173-178 (2018)
3. Sato H, et al. Factors affecting recurrence and prognosis after R0 resection for colorectal cancer with peritoneal metastasis. *J Gastroenterol.* 51,465-72 (2016)
4. Kobayashi H, Kotake K, Kawasaki M, et al. A proposed new Japanese classification of synchronous peritoneal metastases from colorectal cancer: A multi-institutional, prospective, observational study conducted by the Japanese Society for Cancer of the Colon and Rectum. *Ann Gastroenterol Surg.* 7,765-771 (2023)

Q3. Surgical treatment: All patients underwent P/D or eP/D? How about partial pleurectomy which is often performed for thymoma patients with localized pleural dissemination. Were the patients with massive pleural dissemination enrolled into the study? In addition, for patients with de novo Masaoka stage IVa diseases, how did the authors perform surgery (surgical approach etc.)? Was it possible for patients with air leak to perform HITHOC?

Answer: Thank you for your valuable comments. The choice of surgical method was detailed in our protocol article [1]. Minimally invasive (if feasible) or open surgery was conducted on enrolled participants with the primary objective of reducing tumor burden. During the surgery, the surgeons made every effort to remove all visible tumor lesions as completely as possible, selecting the most appropriate surgical approach based on the location and number of the lesions. Partial pleurectomy was

utilized for oligometastatic pleural nodules. Complete pleurectomy was performed selectively, considering the safety of the operation, and was always employed for multiple nodules with extensive dissemination on the parietal pleura. For diaphragmatic dissemination, partial diaphragm resection combined with diaphragm repair was used. In cases where tumor lesions had spread into the pericardium parietal pleura, partial pericardial resection combined with pericardial repair was applied. For nodules in the visceral pleura that were difficult to decorticate, lung wedge resection was often employed. Occasionally, air leaks occur during surgery. In such instances, intraoperative water submersion tests were used to identify the location of the air leak, followed by lung repair via suturing. Pleural decortication was the preferred method, and if necessary, combined resection and repair of invaded tissues were performed.

In this study, patients with massive pleural dissemination were not excluded. For these patients, we endeavored to remove as many metastatic nodules as possible. These patients were associated with high PTI and poor survival after S-HITOC, suggesting that the benefits of S-HITOC are limited for cases with massive pleural dissemination. Therefore, the application of S-HITOC for such patients should be carefully evaluated. We have also emphasized this point in the discussion section.

We hope this response addresses your concerns. Thank you again for your insightful feedback.

Revised in the Manuscript: Page 5-6, line 141-148; Page 8-9, line 233-237; Page 9, line 244-248; Page 12, line 327-331, line 345-349.

Reference:

1. Yang X, et al. Cytoreductive surgery combined with hyperthermic intrathoracic chemotherapy for the treatment of thymic epithelial malignancies with pleural spread or recurrence (CHOICE): a study protocol for a prospective, open, single-arm study. *J Thorac Dis.* 16,760-767 (2024)

Q4. Is HITHOC feasible after pericardial resection? I think longer follow-up (cardiac function etc) should be performed.

Answer: Thank you for your kind reminder. It is widely acknowledged that pericardial defect is a contraindication for HITOC due to the risk of immersing the heart in perfusion liquid and the potential for irreversible damage from chemotherapeutic drugs to the myocardium. In this trial, 8 patients with pericardial lesions in the parietal pleura (not pericardial dissemination) underwent partial pericardial

resection followed by pericardial repair using direct suture or patch filling (**as illustrated in Figure 2 and Supplement Figure 4**). Consequently, these patients were able to receive HITOC on the following day. We observed that patients who underwent pericardial repair did not suffer from severe cardiac function damage during HITOC and follow-up. However, given the small number of patients with pericardial resection and repair in this trial, it is insufficient to draw a very reliable conclusion. Therefore, we will continue to accumulate patients to explore the best treatment for patients with pericardial dissemination. Furthermore, we will follow up with these patients to assess the survival benefit and cardiac function.

In a related study, Klotz and colleagues analyzed the clinical outcomes of 71 patients with malignant pleural mesothelioma (MPM) treated with pleurectomy/decortication (P/D) followed by HITOC with cisplatin and doxorubicin [1]. The pericardium was resected and reconstructed with a xenopericard patch. The peri-operative morbidity was found to be acceptable, with more than 40% of all complications classified as minor and successfully relieved by conservative treatment. The median overall survival was 16.1 months, significantly influenced by histological pattern. Multivariate analysis confirmed that histologic tumor subtype and radicality of resection impact overall survival rates as independent risk factors. This study suggests that HITOC can be safely administered following pericardial repair, although further research is needed to confirm these findings.

We agree with your viewpoint that longer follow-up (cardiac function etc) should be performed. We add the statement and recognize the limitations of our research in revised manuscript.

We hope this response addresses your concerns. Thank you again for your valuable feedback.

Revised in the Manuscript: Page 5-6, line 143-148; Page 11, line 300-302;

Revised in Supplement Figure 4.

Supplementary Figure 4. Typical surgical images during pleurectomy/decortication and extended pleurectomy/decortication, including (A) resection of pleural metastatic nodules, (B) lung wedge resection, (C) partial diaphragmatic resection and repair, and (D) partial pericardiectomy and repair.

Reference:

1. Klotz LV, et al. Multimodal therapy of epithelioid pleural mesothelioma: improved survival by changing the surgical treatment approach. *Transl Lung Cancer Res.* 11,2230-2242 (2022)

Q5. The follow-up period is also insufficient to evaluate the treatment effectiveness in thymoma because of its slow progression. I think patients with R2 resection should be excluded for evaluation of progression free survival (PFS).

Answer: Thank you for your kind reminder. We acknowledge that the follow-up period for this trial is indeed relatively short, which is one of the limitations of this study, as we have mentioned in the “Limitations” section. However, it is important to note that the primary focus of this trial is to evaluate the perioperative outcomes and short-term survival of S-HITOC, to provide readers with an understanding of the safety and feasibility of this treatment regimen. Our team is committed to continuing the follow-up of these patients and looks forward to reporting on their long-term survival.

Although thymic tumors are currently considered to be relatively indolent and to progress more slowly than other thoracic malignant tumors, advanced thymic tumors can still exhibit disease progression. There are two main reasons for including R2 patients in the evaluation of progression-free survival (PFS). Firstly, R2 patients often experience tumor progression in a short time, allowing us to assess the efficacy of S-HITOC for thymic tumors within a limited timeframe. Secondly, the

resectability of tumors can be inconsistent between preoperative and intraoperative evaluations. Excluding all R2 patients might result in an excessive loss of valuable clinical information, considering the rarity of this disease. Therefore, it is still necessary and significant to include R2 patients in this trial.

For patients with pleural dissemination, due to the possibility of potentially invisible metastatic lesions, theoretically, complete resection might not be achievable. In this trial, we adopted the surgical assessment criteria from ovarian cancers with peritoneal metastases, namely: R0, complete cytoreductive surgery without residual visible disease; R1, optimal cytoreductive surgery with residual tumors measuring no more than 10 mm; R2, incomplete cytoreductive surgery with residual lesions measuring > 10 mm in diameter. This standard helps us to objectively describe the surgical resection. Based on these criteria, we calculated the PFS of all patients, as shown in Figure 3A. Among all enrolled patients, the estimated 1- and 2-year PFS rates were 97.3% and 82.8%, respectively.

Furthermore, incomplete resection typically indicates a heavier tumor burden, which is often associated with worse survival outcomes. Therefore, we compared the survival differences between patients who received R0/1 resection and those who received R2 resection, as shown in Figure 3D. The results indicate that, compared to those with R2 resection, patients with R0/1 resection had statistically better PFS, although the difference was not significant. A longer follow-up period is needed to draw more definitive conclusions.

We hope this explanation addresses your concerns. Thank you for your attention and feedback.

Figure 3. Survival analysis of (A) PFS among all patients; (B) OS among all patients; (C) PFS based on the PTI score; (D) PFS based on the surgical resection completeness; (E) PFS based on the pleural dissemination classification; (F) PFS based on the tumor stage.

Revised in the Manuscript: Page 5, line 120-122; Page 5-6, line 145-148; Page 7, line 192-194;

Page 11, line 302-304, line 319-321; Page 12, line 345-349;

Revised in Supplement Table 2; Figure 3.

Q6. The S-HITOC treatment for thymic tumor is performed for patients with massive pleural disseminations. However in this paper, patients with high PTI scores had worse PFS. What kind of patients with thymic epithelial tumors can we use this procedure for?

Answer: Thank you for your valuable question. Most surgeons and clinicians have indeed attempted to apply S-HITOC treatment for thymic tumors with pleural dissemination. However, due to the rarity of stage IVa TETs and the limited experience in this area, there has been no consensus on which specific TETs with pleural dissemination would benefit most from the S-HITOC procedure. When designing the study protocol, we did not impose restrictions on the specific features of pleural metastases during the screening process. The only exclusions were for TET patients who had intolerable treatment related risks. Consequently, our trial included both patients with extensive pleural metastasis and those with oligopleural metastasis. This approach allows us to observe the differences in the safety and efficacy of S-HITOC across patients with varying tumor burdens and to analyze the best indications for these heterogeneous patients.

In a study by Yellin et al., the mean overall survival (OS) after S-HITOC in patients with TETs with pleural spread was reported to be 12 years, with 5- and 10-year progression-free survival (PFS) rates of up to 47.6% and 17.9%, respectively [1]. Maury et al. reported that after S-HITOC, patients with pleural recurrences of thymomas had a mean OS time of 63 months and a mean local disease-free interval time of 41 months [2]. Aprile et al. showed that an OS time of up to 64 months and a PFS time of up to 53 months could be achieved in patients with thymoma pleural relapses after S-HITOC [3]. These studies indicate that S-HITOC is an effective protocol for the treatment of TETs with pleural spread or recurrence.

To formally and objectively assess the location and severity of pleural metastasis, we introduced the concept of the Pleural Tumor Index (PTI). Our study findings indicate that patients with a low PTI (less than 10) had a higher survival rate after undergoing S-HITOC compared to those with a high PTI (more than 10). This suggests that patients with a PTI of less than 10 may be better candidates for benefiting from S-HITOC. It is worth noting that patients with extensive pleural metastasis typically have a high PTI and often cannot undergo complete resection. Previous literature has shown that the survival outcomes for these patients remain poor even after receiving traditional treatments such as chemotherapy or chemoradiotherapy. In our trial, the efficacy of S-HITOC in patients with a high PTI was not as satisfactory as in those with a lower PTI. However, it cannot be denied that S-HITOC may

still represent a treatment option for TETs patients with pleural spread or recurrence.

We hope this explanation addresses your concerns. Thank you for your attention and feedback.

Revised in the Manuscript: Page 7, line 187-194; Page 8-9, 233-239; Page 11, line 302-304; Page 12, line 339-349.

Reference

1. Yellin A, et al. Resection and heated pleural chemoperfusion in patients with thymic epithelial malignant disease and pleural spread: a single-institution experience. *J Thorac Cardiovasc Surg.* 145,83-7 (2013)
2. Maury JM, et al. Intra-thoracic chemo-hyperthermia for pleural recurrence of thymoma. *Lung Cancer.* 108,1-6 (2017)
3. Aprile V, et al. Surgical treatment of pleural recurrence of thymoma: is hyperthermic intrathoracic chemotherapy worthwhile? *Interact Cardiovasc Thorac Surg.* 30(5):765-772 (2020)

Q7. For patients with thymoma, recurrence does always not lead to death and deaths are often not due to recurrence.

Answer: Thank you for your valuable comment. It is indeed true that thymoma is generally considered a relatively indolent malignant tumor, as you pointed out. However, for advanced thymic tumors, particularly those that are unresectable, disease progression and related deaths are not uncommon. In this trial, out of the enrolled patients, two patients (4.4%) passed away before the latest follow-up visit; one patient succumbed to tumor progression in the 34th month after treatment, and the other to an infectious disease in the 35th month post-treatment. The 1- and 2-year overall survival (OS) rates were 100.0% (100.0%-100.0%). These results underscore the positive impact of S-HITOC in the treatment of advanced thymic tumors.

Several large-sample national databases have demonstrated that early-stage thymic tumor patients typically exhibit satisfactory survival rates, with deaths often unrelated to recurrence [1,2]. This suggests that aggressive tumor removal is crucial for improving patient survival outcomes upon the discovery of thymic tumors. Furthermore, our previous cohort study, published in the *Journal of Thoracic and Cardiovascular Surgery (JTCVS)*, indicated that for locally advanced thymic epithelial tumors (TETs), surgery (including both minimally invasive and open procedures) is associated with

good survival outcomes [3]. These encouraging findings prompted us to explore whether surgery could be a viable option for potentially resectable stage IV thymic tumors, such as those with resected pleural nodules.

To date, numerous surgeons have reported on the use of S-HITOC for the treatment of advanced thymic tumors. However, these reports have been largely anecdotal, with small sample sizes and retrospective designs. Therefore, we initiated this trial to investigate the safety and efficacy of a standardized S-HITOC protocol for TETs with pleural dissemination. To our knowledge, this is the first study to assess the uniform application of S-HITOC in TETs with pleural involvement. Our findings revealed that post-treatment complications were manageable, with no increased need for reoperation or risk of postoperative mortality. The median post-treatment hospital stay was not significantly prolonged. Most patients did not experience intolerable pain following treatment and regained their quality of life within two months. The latest follow-up data are encouraging, with recurrence observed in 5 out of 45 patients (11.1%) and death in 2 out of 45 patients (4.4%). The 2-year progression-free survival (PFS) and OS rates were 82.8% and 100.0%, respectively. Thus, our study demonstrates that S-HITOC is both safe and effective for patients with TETs exhibiting pleural spread or recurrence.

By strictly adhering to the S-HITOC protocols, the trial outcomes are more objective and reliable, providing readers with a comprehensive understanding of this treatment approach.

Thank you for your attention and feedback.

Revised in the Manuscript: Page 7, line 202-205; Page 8, line 213-222.

Reference:

1. Ruffini E, et al; Members of the IASLC Staging and Prognostic Factors Committee and of the Advisory Boards, and Participating Institutions. The International Association for the Study of Lung Cancer Thymic Epithelial Tumors Staging Project: Proposal for a Stage Classification for the Forthcoming (Ninth) Edition of the TNM Classification of Malignant Tumors. *J Thorac Oncol.* 18,1655-1671 (2023)
2. Yin Y, et al. Investigating the impact of tumor size on survival outcomes in thymoma and thymic carcinoma patients using the SEER database. *Sci Rep.* 14,27680 (2024)
3. Yang X, et al. Perioperative outcomes and survival of modified subxiphoid video-assisted thoracoscopic surgery thymectomy for T2-3 thymic malignancies: A retrospective comparison study. *J Thorac Cardiovasc Surg.* 168,1550-1559 (2024)

Response to Reviewer #2 (Thoracic Surgery, clinical)

Dear authors, thank you for giving me the opportunity to review your article. The study investigated cytoreductive surgery combined with hyperthermic intrathoracic chemotherapy in treating stage IV thymic epithelial tumors with pleural spread or recurrence. You reported a 2-year progression-free survival rate of 82.8% and a 100% overall survival rate. The treatment demonstrated acceptable levels of complications, manageable pain and post-operative quality of life, and an important remission rate for patients with myasthenia gravis. You developed the Pleural Tumor Index which was correlated with PFS.

This is the first prospective, single-arm phase II study that could potentially establish S-HITOC as a future standard care option.

Answer: Thank you very much for taking the time to review our study. We truly appreciate your constructive comments and the recognition of our work. The design of our experiment was a result of multiple, detailed, and comprehensive discussions with our entire multidisciplinary team (MDT). We ensured that the protocol was implemented objectively and rigorously throughout the study.

Thanks to the concerted efforts of the team, we were able to complete the enrollment and treatment of patients within just 2 years. We hope that our experience and the data we have gathered will provide readers with a deeper understanding of this disease and the treatment approach involving cytoreductive surgery combined with hyperthermic intrathoracic chemotherapy.

Once again, we express our sincere gratitude for your guidance and recognition of our work. Thank you.

Some comments:

Q1. I should include the WHO classification of the thymoma and correlate with survival.

Answer: Thank you for your insightful comments. In response to your suggestions, we have incorporated the WHO classification of thymoma into Table 1. This addition is intended to provide patients with a clearer understanding of the characteristics of our patient cohort, as detailed below.

Six patients were thymic carcinoma. Among the 39 thymoma patients included in our study, the distribution according to the WHO classification is as follows: 1 patient (2.6%) was classified as type A, 1 patient (2.6%) as type AB, 3 patients (7.7%) as type B1, 14 patients (35.9%) as type B2, 10 patients (25.6%) as a mix of B2/B3, and 10 patients (25.6%) as type B3.

We believe that this modification will enhance the comprehensibility of our data and contribute to a more thorough understanding of the patient demographics in our study.

Table 1. Baseline Characteristics (part.)

Pathology	No. (%)
Thymoma	39 (86.7)
A	1 (2.2)
AB	1 (2.2)
B1	3 (6.6)
B2	14 (31.1)
B2/B3 mix	10 (22.2)
B3	10 (22.2)
Thymic carcinoma	6 (13.3)

In this trial, we observed that the 1-year and 2-year progression-free survival (PFS) rates of stage IV thymoma patients were 97.0% (91.3%-100.0%) and 87.3% (70.4%-100.0%), respectively. For stage IV thymic carcinoma patients, the 1-year and 2-year PFS rates were 100.0% (100.0%-100.0%) and 66.7% (30.0%-100.0%), respectively. Notably, neither thymoma nor thymic carcinoma patients experienced postoperative mortality within the 2-year follow-up period. Due to the limited sample size, we did not find a significant difference in PFS or overall survival (OS) between the different pathological classifications of thymic tumors. In accordance with your suggestion, we have added the survival plot to better illustrate these findings in the revised manuscript.

We believe that this addition will enhance the visual presentation of our data and provide a clearer understanding of the survival outcomes across different patient groups.

Supplement Figure 10. Survival difference of patients in two groups with two pathologic types of thymic epithelial tumors. A, Progression-free survival; B, Overall Survival.

Revised in the Manuscript: Page 5, line 124-125; Page 7, line 187-189, line 200-201, line 204-205;

Revised in Table 1; Supplement Table 2; Supplement Figure 10.

Q2. Why did you include thymic carcinoma on your study? This is clearly not the same disease than thymoma.

Answer: Thank you for your question. You are correct that thymoma and thymic carcinoma are distinct diseases, with thymic carcinoma generally being more aggressive and prone to progression or disease-related death compared to thymoma, even at early stages. However, the difference in progression and disease-related death between thymoma and thymic carcinoma with pleural dissemination has not been well established. By including thymic carcinoma in our study, we were able to observe the efficacy of S-HITOC in thymic carcinoma and analyze the survival and perioperative outcomes differences between thymoma and thymic carcinoma with pleural dissemination through subgroup analyses. Thymic carcinoma is also a heterogeneous disease, Biological behavior of thymic carcinoma with pleural dissemination is varied from that of patients with hematogenous metastasis. A huge difference exist in thymic carcinoma cases between pleural dissemination and hematogenous metastasis, which is in relation to clinicopathological features, treatment, tumor response and prognosis. It is certain that tumor reaction of thymic carcinoma with pleural dissemination to S-HITOC is quite distinct from that of patients with hematogenous metastasis. We believe this is a topic worthy of exploration.

It is also important to note that thymic epithelial tumors are relatively rare, and the number of

patients with pleural dissemination is quite limited. If we had restricted the pathological types at the time of enrollment, we would have excluded many patients, leading to a prolonged enrollment period. Therefore, based on practical considerations, we did not impose limitations on the pathological types of thymic epithelial tumors in this trial.

In this trial, we attempted to compare the prognostic differences between the two pathological types. Based on the current sample size and follow-up duration, our findings are as follows: For stage IV thymoma patients, the 1-year and 2-year progression-free survival (PFS) rates were 97.0% (91.3%-100.0%) and 87.3% (70.4%-100.0%), respectively. For stage IV thymic carcinoma patients, the 1-year and 2-year PFS rates were 100.0% (100.0%-100.0%) and 66.7% (30.0%-100.0%), respectively. Neither thymoma nor thymic carcinoma patients experienced postoperative mortality within the 2-year follow-up period. Due to the limited sample size, we did not find a significant difference in PFS or overall survival (OS) between the two WHO pathological classifications of thymic tumors (Supplement Figure 10).

In the future, we plan to increase the sample size and conduct further follow-up to explore the prognostic differences between these two pathological types more thoroughly. We hope this explanation addresses your concerns.

Supplement Figure 10. Survival difference of patients in two groups with two pathologic types of thymic epithelial tumors. A, Progression-free survival; B, Overall Survival.

Revised in the Manuscript: Page 7, line 187-189, line 200-201, line 204-205; Table 1; Supplement

Table 2;

Revised in Supplement Figure 10.

Q3. Generally speaking, a thymoma is an indolent tumour with slow growing recurrence. In my opinion, OS or even PFS are difficult to interpret for this particular tumour. You should need to publish late results. For me, it is difficult to make any conclusion without comparative groups, because after 1 year (your median follow-up), I would be sure that the results would be similar without or without HITOC. This point should be explained on the title (initial peri-operative outcomes or morbidity of such procedure)

Answer: Thank you for your question. Your insight is both valuable and reflective of the current challenges in this field of research. Over the past decade, only a handful of studies have reported on the use of S-HITOC for the treatment of thymic tumors with pleural dissemination. The regional nature and rarity of these cases have led to retrospective studies with small sample sizes. Surgeons often determine the timing, temperature, and treatment schedule based on their individual experience, making it difficult to establish a consensus on this treatment approach. It is against this backdrop that we designed our trial.

Our study represents the first prospective trial with a standardized S-HITOC treatment strategy aimed at exploring its application in the treatment of thymic tumors with pleural dissemination. The primary focus of this study is to assess the safety and feasibility during the perioperative period. Consequently, we have used the complication rate as the primary endpoint and prognosis as the secondary endpoint.

Our findings indicate that post-treatment complications were within acceptable limits, with no increased need for reoperation or risk of postoperative mortality. The overall median post-treatment hospital stay was not significantly prolonged. Most patients did not experience intolerable pain following treatment and regained their quality of life within two months. The latest follow-up data are encouraging, with recurrence observed in 5 out of 45 patients (11.1%) and death in 2 out of 45 patients (4.4%). The 2-year progression-free survival (PFS) and overall survival (OS) rates were 82.8% and 100.0%, respectively. Thus, our study demonstrates that S-HITOC is both safe and effective for TETs exhibiting pleural spread or recurrence.

We acknowledge that randomized controlled trials (RCTs) are the gold standard for evaluating the efficacy of interventions. Based on our preliminary results, we have completed the study design for a multicenter RCT and plan to further explore the feasibility of this treatment option in a more in-depth, objective, and rigorous manner. In response to your suggestion, we have revised the title to:

“Short-term Outcomes of Cytoreductive Surgery and Hyperthermic Intrathoracic Chemotherapy for Thymic Epithelial Tumors with Pleural Spread or Recurrence: A Prospective, Single-Arm, Phase II Study”. This change emphasizes that the focus of this article is on short-term results. Additionally, in the limitations section, we have reiterated that long-term OS requires further investigation.

Thank you for your kind reminder and continued support.

Revised in the Manuscript: Page 1, line 1-3; Page 2, line 42-44; Page 3, line 60-62, Page 4, line 112-114; Page 8, line 213-222; Page 11, line 304-306.

Q4. I like the idea of PTI score's prognostic value. This is a good idea to better compare the patients that are quite heterogeneous.

Answer: I really appreciate your recognition of our work. In fact, due to the rarity of the case, even though our center is one of the national regional healthcare centers, our recruitment work is still very difficult. In clinical practice, we have found that even patients with pleural dissemination have significantly different disease burdens. Reviewing domestic and international guidelines, there is currently no unified standard to objectively describe this clinicopathological manifestation [1,2]. This is also the original intention of the proposal of the concept “pleural tumor index (PTI)”, that is, to objectively and comprehensively describe the severity of pleural dissemination by designing a data-based and systematic indicator to facilitate clinicians in disease diagnosis and treatment. Interestingly, we found that patients with PTI scores >10 had worse PFS (HR=18.26; $p<0.001$) than those with PTI scores ≤ 10 (Figure 3C), suggesting that low-PTI thymic tumor patients could try S-HITOC as the first-line treatment choice. For these high PTI (more than 10) patients, additional effective treatment options may need to be explored in the future. Based on the current results, we believe that PTI may become an important tool for clinicians in the assessment and treatment for thymic tumors with pleural dissemination, and we will further explore its feasibility in the randomized controlled trial. Thank you again for your recognition of our work.

Revised in the Manuscript: Page 11, line 302-304, Page 12, line 337-345.

References:

1. Fang W, et al. China Anti-Cancer Association Guidelines for the diagnosis, treatment, and follow-up

of thymic epithelial tumors (2023). *Mediastinum*. 8,27 (2024)

2. NCCN Clinical Practice Guidelines in Oncology. Thymomas and Thymic Carcinomas. 2021; Version 1 (2021)

Q5. Sufficient detail is provided for reproducibility, including precise dosages, chemotherapy temperatures, and surgical techniques.

Answer: I truly appreciate your recognition of our work. The primary objective of our research is to treat thymic tumor patients with pleural dissemination using a standardized S-HITOC process, thereby assisting clinicians in understanding this treatment approach. Thank you for your attention to our trial design. The protocol for this trial was meticulously crafted based on our primary exploratory clinical experience and a thorough review of previous studies [1-4]. It was not developed in isolation but rather through an open and collaborative process. We engaged with a multidisciplinary team of experts, including surgeons, oncologists, radiologists, and pathologists, to ensure that the protocol was comprehensive and scientifically sound. The study protocol had been published [5]. Our approach was to build upon the existing body of knowledge and our clinical insights to design a study that would provide meaningful and actionable data. We believe that this collaborative and evidence-based approach has been instrumental in shaping the robustness and relevance of our trial protocol.

To carry out this study, we designed the trial protocol a year in advance and established both a multidisciplinary team (MDT) for research and a dedicated follow-up team. Once the trial commenced, we administered the standardized treatment to each enrolled patient and meticulously objectively recorded the data. Our experience has shown that S-HITOC is effective and safe for treating thymic tumor patients with pleural dissemination. We also sincerely hope that more medical professionals will pay attention to this rare patient population and consider the S-HITOC regimen as a viable treatment option.

Thank you once again for your support and recognition.

Revised in the Manuscript: page 12, line 327-336.

Reference

1. Wang S, et al. Induction Therapy Followed by Surgery for Unresectable Thymic Epithelial Tumours. *Front Oncol*. 11,791647 (2022)
2. Maury JM, et al. Intra-Thoracic Chemo-Hyperthermia for pleural recurrence of thymoma. *Lung*

Cancer. 108,1-6 (2017)

3. Monneuse O, et al. Long-term results of intrathoracic chemohyperthermia (ITCH) for the treatment of pleural malignancies. *Br J Cancer*. 88,1839-43 (2003)
4. Aprile V, et al. Surgical treatment of pleural recurrence of thymoma: is hyperthermic intrathoracic chemotherapy worthwhile? *Interact Cardiovasc Thorac Surg*. 30,765-772 (2020)
5. Yang X, et al. Cytoreductive surgery combined with hyperthermic intrathoracic chemotherapy for the treatment of thymic epithelial malignancies with pleural spread or recurrence (CHOICE): a study protocol for a prospective, open, single-arm study. *J Thorac Dis*. 16,760-767 (2024)

Response to Reviewer #3 (lung cancer therapy)

Firstly, I would like to congratulate the authors on their efforts to study a relatively large series of patients with this rare condition.

Secondly, this study does add to the existing literature on this subject.

I have the following comments which I would ask the authors to address:

Answer: Thank you for your kind words and for acknowledging our efforts in studying this rare condition. We are indeed grateful for your positive feedback and recognition of the contribution our study makes to the existing literature. We are more than happy to address any comments or questions you may have. Please feel free to share your specific concerns, and we will provide detailed responses to ensure that your feedback is fully considered and addressed.

Thank you once again for your time and valuable input.

Q1. As this was a prospective study, in the same time frame how many were assessed for this treatment but were not found to be suitable and why was that?

Answer: Thank you for your suggestion. We apologize for not emphasizing the screening flow chart in Supplement Figure 3 in the primary draft. We understand the importance of clearly presenting the patient enrollment process, especially given the rarity of the condition and the challenges in recruitment.

To address your comment, we have revised the manuscript to include a detailed patient screening flow chart. As you pointed out, due to the rarity of the cases, patient enrollment was relatively slow, even though our center is one of the largest national healthcare centers. During the study period, we screened a total of 53 patients who might have been eligible. Of these, 2 patients were confirmed to have mesothelioma by biopsy, 1 patient was found to have liver metastases, and 2 patients refused to register. Among the 48 patients who met the inclusion criteria, 2 patients were found to have a history of other malignant tumors, and 1 patient refused to receive the S-HITOC treatment. Consequently, 45 patients were finally enrolled, and all 45 patients completed the planned S-HITOC regimen.

Due to word limit in the main text, we have uploaded the patient screening flow chart as an attachment (Supplementary Figure 3) in this revised manuscript. We believe this addition will provide a clearer understanding of the patient selection process and enhance the transparency of our study. Thank you again for your reminder and valuable feedback.

Supplementary Figure 3. The screening flowchart in this study.

Revised in the Manuscript: Supplement Figure 3.

Q2. I note the high number of patients (48.9%) who had residual tumour left after surgery. Was the tumour thought to be fully resectable in these patients preoperatively or was the intent just debulking?

Answer: Thank you for your thoughtful question and your careful attention to the details of our surgical approach. We appreciate the opportunity to clarify our methodology.

The primary aim of our surgery was to achieve the most extensive resection of visible metastatic lesions while prioritizing patient safety and surgical feasibility. We did not have any cases where the intention was solely debulking; rather, the goal was always to maximize tumor removal, ensuring the greatest possible extent of resection while minimizing surgical risk.

For stage IVa thymic tumors with pleural dissemination, complete resection is often theoretically unattainable, primarily due to the presence of microscopic disease or invisible lesions that are not detectable at the time of surgery. In such instances, we aimed to resect as much visible tumor as possible. When complete resection was not feasible due to the extent of pleural involvement, the presence of invisible lesions, or massive visceral pleural metastasis, we performed optimal cytoreductive surgery, striving to achieve the best possible outcome given the constraints.

To quantify the extent of surgical resection, we used the following classification criteria: R0: Complete resection with no residual visible disease; R1: Optimal resection with residual disease no larger than 10 mm; R2: Incomplete resection with residual lesions larger than 10 mm. These criteria provided an objective framework for evaluating the success of our surgical approach. While complete resection is more feasible in cases of pleural oligometastasis, it becomes increasingly challenging as the extent of pleural metastasis increases. In cases of massive pleural involvement, achieving complete resection of all visible disease can be unrealistic, thus we performed debulking to the extent possible, balancing the need for disease control with the preservation of patient safety.

Additionally, current imaging modalities, particularly enhanced CT, do not always provide an accurate assessment of pleural dissemination. Our study highlights the fact that the extent of pleural dissemination is frequently underestimated by imaging, which contributed to the presence of residual disease in nearly 49% of patients following surgery. Importantly, S-HITOC plays a crucial role in addressing these residual lesions and invisible metastases. Our survival data, which show favorable progression-free survival (PFS) and overall survival (OS) during follow-up, further support the efficacy of HITOC in managing these residual tumor burdens.

In light of the challenges associated with incomplete resection, especially in cases of extensive pleural metastasis, we introduced the concept of the pleural tumor index (PTI). The PTI is a quantitative, data-driven tool designed to assess the severity and distribution of pleural dissemination, providing a more objective measure of pleural tumor burden that could assist in clinical decision-making. We observed that patients with a lower PTI (≤ 10) had significantly better survival outcomes following S-HITOC compared to those with a higher PTI (> 10). This finding suggests that the PTI could be a valuable tool in identifying patients who may benefit most from S-HITOC. Based on these promising results, we believe that the PTI could become an important addition to clinical practice in the evaluation and treatment of thymic tumors with pleural dissemination. Moving forward, we plan to validate this index in a randomized controlled trial to further explore its potential as a robust and clinically applicable tool.

Thank you again for your insightful feedback, and we hope this clarification enhances the understanding of our surgical approach and the role of S-HITOC in treating TETs with pleural dissemination.

Thank you again for your recognition of our work.

Revised in the Manuscript: Page 5, line 136-138; Page 5-6, line 143-148; Page 9, line 252-255; Page 11, line 296-299, line 302-304; Page 12, line 337-349.

Q3. Can you describe the surgical technique in more detail? Why was tumour left in nearly half of the patients? Was there an intentional limit to your extent of surgery?

Answer: We sincerely apologize for the initial manuscript's insufficient description of the surgical procedures. A more detailed account of the surgical approach was indeed provided in our protocol article, which have been published [1]; however, to further clarify the methodology used in this trial, we would like to offer a more comprehensive description of the surgical techniques employed.

Surgical Approach

In this trial, all participants underwent minimally invasive surgery when feasible, or open surgery when necessary, with the primary objective of reducing tumor burden. Every effort was made to remove all visible tumor lesions as completely as possible. The choice of surgical method was determined based on the location and number of lesions, and the specific surgical techniques employed included: partial pleurectomy: applied for patients with oligometastatic pleural nodules; complete pleurectomy: performed selectively, based on the safety of the procedure, and indicated for patients with multiple nodules involving extensive parietal pleural dissemination; partial diaphragm resection with repair: used for diaphragmatic involvement; partial pericardial resection with repair: applied to patients with tumors extending into the pericardium; lung wedge resection: performed for nodules on the visceral pleura that were difficult to decorticate.

Surgical Extent

Our approach did not impose limits on the extent of surgery. While some centers may advocate for extrapleural pneumonectomy (EPP) in cases of massive visceral pleural metastasis, this technique was not deliberately avoided, but indeed not performed in our trial, due to the substantial risks associated with significant pulmonary function reduction and potential severe complications.

Trial Objective

The primary objective of this trial was to evaluate the combined efficacy of cytoreductive surgery followed by hyperthermic intrathoracic chemotherapy (S-HITOC) in patients with TETs exhibiting pleural spread or recurrence. Cytoreductive surgery was performed to remove as much visible tumor as

possible, and to minimize residual lesions. Following surgery, HITOC was administered to target residual disease and microscopic metastases, aiming to enhance disease control and improve patient outcomes.

In summary, our surgical approach was comprehensive and tailored to each patient's specific clinical condition. The goal was to achieve maximal tumor removal while safeguarding patient safety. The combination of cytoreductive surgery and S-HITOC was designed to address both visible and microscopic disease, providing a multifaceted approach to improving outcomes in TETs with pleural dissemination.

We hope this detailed explanation sufficiently addresses your concerns. Thank you for your thoughtful feedback and for your continued understanding.

Revised in the Manuscript: Page 5, line 136-138; Page 5-6, line 143-148; Page 9, line 252-255; Page 11, line 296-299, line 302-304; Page 12, line 337-349.

Reference:

1. Yang X, et al. Cytoreductive surgery combined with hyperthermic intrathoracic chemotherapy for the treatment of thymic epithelial malignancies with pleural spread or recurrence (CHOICE): a study protocol for a prospective, open, single-arm study. *J Thorac Dis.* 16,760-767 (2024)

Q4. Can you clarify your definition of R1 as <1cm residual tumour? Where is this definition from?

Answer: We apologize for the unclear and insufficient description of the resection definition in the initial manuscript. We greatly appreciate your valuable feedback and would like to provide a more comprehensive explanation to clarify our approach.

Cytoreductive surgery is designed to reduce the tumor burden, particularly in cases of intraabdominal spread, such as peritoneal carcinoma. It is a well-established procedure in the treatment of ovarian cancer, but it is also applicable to other abdominal and thoracic malignancies or metastases. In the context of advanced ovarian cancer with peritoneal metastasis, a complete resection is defined as the absence of a macroscopically visible tumor after surgery. If visible tumors remain, they are classified according to their largest diameter:

Optimal Debulking: Residual tumors with the largest diameter ≤ 1 cm.

Suboptimal Debulking: Residual tumors with the largest diameter > 1 cm.

The prognostic value of complete and optimal debulking has been well established in the literature, with numerous studies confirming its impact on patient outcomes [1]. A prospective randomized trial has shown that optimal debulking significantly improves survival [2]. Additionally, an exploratory analysis combining three prospective phase 3 trials categorized surgical outcomes as follows:

Group A: Complete resection with no visible residual disease.

Group B: Small residuals (1-10 mm).

Group C: Residual tumor >10 mm.

These results demonstrated a stepwise decline in prognosis as residual tumor size increased, with tumors exceeding 10 mm having the poorest outcomes. Consequently, cutoff values of 10 mm have become standard for distinguishing the extent of resection in these settings.

The pattern of pleural metastasis in thymic epithelial tumors (TETs) shares similarities with the peritoneal metastasis seen in advanced ovarian cancer. As such, we adopted the same resection grading system to assess the extent of resection in TETs with pleural dissemination. The resection grading criteria are as follows:

R0: Complete cytoreductive surgery with no residual visible disease.

R1: Optimal cytoreductive surgery with residual tumors measuring ≤ 10 mm.

R2: Incomplete cytoreductive surgery with residual tumors > 10 mm.

This grading system provides an objective framework for evaluating the extent of tumor removal and enables consistent assessment across cases.

We hope this explanation provides greater clarity regarding our approach to resection assessment in this trial. By employing this standardized resection grading system, we aim to enhance the objectivity and comparability of our results, ensuring more robust and reproducible findings. Thank you again for your constructive feedback, and we hope this response addresses your concerns.

Revised in the Manuscript: Page 5-6, line 141-148; Page 12, line 345-349.

Reference

1. Bristow RE, et al. Survival effect of maximal cytoreductive surgery for advanced ovarian carcinoma during the platinum era: a meta-analysis. *J Clin Oncol.* 20,1248-1259 (2002)
2. van der Burg ME, et al. The effect of debulking surgery after induction chemotherapy on the prognosis in advanced epithelial ovarian cancer. Gynecological Cancer Cooperative Group of the

European Organization for Research and Treatment of Cancer. N Engl J Med. 332,629-634 (1995)

3. du Bois A, et al. Role of surgical outcome as prognostic factor in advanced epithelial ovarian cancer: a combined exploratory analysis of 3 prospectively randomized phase 3 multicenter trials: by the Arbeitsgemeinschaft Gynaekologische Onkologie Studiengruppe Ovarialkarzinom (AGO-OVAR) and the Groupe d'Investigateurs Nationaux Pour les Etudes des Cancers de l'Ovaire (GINECO). Cancer. 115,1234-44 (2009)

Q5. Can the authors be more specific in the future selection of patients for this invasive technique? Are there clear exclusion criteria from their data and experience?

Answer: Thank you for your insightful question. You are correct in noting that thymic tumors with pleural dissemination are relatively rare, and we are still in the process of accumulating clinical experience in treating this condition. As a result, we did not proceed directly to a phase III clinical trial. Instead, we initiated this exploratory phase II study to assess the safety and short-term survival outcomes of our treatment regimen.

From this trial, we have derived several important observations: safety: our treatment regimen has been demonstrated to be safe and well-tolerated; postoperative quality of life: the treatment significantly improves postoperative quality of life, which is a critical aspect of patient care; short-term prognosis: the regimen has a positive impact on short-term survival outcomes, providing initial evidence of its efficacy. We also found pleural tumor index (PTI) could be used as a prognostic indicator. Patients with a low PTI (≤ 10) exhibit significant benefits from S-HITOC, suggesting that it may serve as a first-line treatment for this subgroup. Patients with a high PTI (> 10) seem to derive less benefit from the current regimen, indicating that alternative treatment strategies should be explored for these individuals.

Building on the findings of this phase II study, we believe that the PTI could become an important tool for clinicians in assessing and treating thymic tumors with pleural dissemination. As such, we plan to investigate its feasibility further in a randomized controlled trial (RCT). In subsequent trials, we will compare S-HITOC to traditional treatment options in patients with PTI less than 10 to determine the most effective therapeutic approach. For patients with PTI greater than 10, we will continue to explore alternative treatment strategies, as the current regimen may not be as beneficial for this subgroup.

We are grateful for your recognition of our work. The findings from this study provide a solid

foundation for further research and the clinical management of thymic tumors with pleural dissemination. We are committed to advancing our understanding of this rare condition and look forward to sharing the results of future trials to improve treatment options.

Thank you again for your valuable feedback and continued support.

Response to Reviewer #4 (Biostats)

This trial evaluated outcomes of cytoreductive surgery with hyperthermic intrathoracic chemotherapy (S-HITOC) for thymic epithelial tumors (TETs) with pleural spread or recurrence. Between August 2021 and February 2024, 45 patients underwent surgery followed by HITOC with doxorubicin and cisplatin. Complete resection was achieved in 51.1% of cases, with a Clavien-Dindo grade ≥ 3 complication rate of 17.8%. At a median follow-up of 12.2 months, 2-year progression-free survival (PFS) and overall survival (OS) were 82.8% and 100%, respectively. Patients with a pleural tumor index (PTI) >10 had significantly worse PFS. Although HITOC demonstrated promising oncological benefits and effective myasthenia gravis symptom control, several issues remain regarding study design and conclusions.

Q1.1 As a single-center, small-sized study, caution is necessary when interpreting these findings. The sample size is justified by a reduction in treatment-related adverse events (AEs) from 30% to 15%, but several questions arise. How are surgery-related AEs defined? Is this specific to certain AEs or pooled together?

Answer: Thank you for your suggestion. In this study, we used pooled treatment-related adverse events. In this trials, treatment-related adverse events are defined as any undesirable or harmful effects that occur as a result of the intervention being studied (such as a drug, surgery, or therapy). Due to the short-time interval between surgery and HITOC, surgery-related adverse events and HITOC-related adverse events were not distinguished in this study.

Key points for the definition of treatment-related adverse events in this trial:

- (1) Causality: treatment-related adverse events are directly attributed to the treatment and not due to underlying conditions, pre-existing diseases, or other external factors.
- (2) Severity: All types and severity of adverse events that are related to treatment will be recorded. Treatment-related adverse events will be stratified according to the Common Terminology Criteria for Adverse Events, Version 5.0 (CTCAE v5.0) and Clavien-Dindo Classification. Major treatment-related adverse events were defined as those with grade ≥ 3 .
- (3) Onset: treatment-related adverse events may occur at any point during the treatment period, including during treatment administration, immediately after, or even after treatment has been completed. In this trial, all treatment-related adverse events were collected continuously during the treatment period and for a minimum of 90 days after surgery.

(4) Examples: These can include symptoms like nausea, vomiting, fatigue, rash, and organ toxicity, or more serious outcomes such as cardiovascular events, severe infections, or life-threatening conditions.

Thank you for your question. We hope this explanation helps clarify the definitions and importance of treatment-related adverse events in clinical trials.

Revised in the Manuscript: Page 6, line 151-161; Page 11, line 277-284; Page 13, line 363-365.

Q1.2 No formal test against the 30% threshold is presented, raising concerns about the relevance of this sample size justification for the primary analysis.

Answer: Thanks for your kinder suggestion. The selection of specific thresholds for major treatment-related adverse events in clinical trials—such as 30% and 15%—is typically based on several factors, including safety considerations, clinical relevance, statistical power, and previous clinical data. In our sample size calculation, A major treatment-related complication rate within 15% was considered manageable and a rate greater than 30% was considered unsafe. Therefore, 37 patients were required at a one-sided significance level of 0.1, with 80% power to detect a 15% difference in the major treatment-related adverse events rate. In this study, the major treatment-related adverse events rate was 17.8%, and the upper limit of the one-sided 90% confidence interval was 27.3%. When the binomial exact test was used to test against the 30% threshold, the one-sided P value was 0.051, indicating that our study has reached the criteria of success.

Thank you again for your constructive feedback, and we hope this response addresses your concerns.

Revised in the Manuscript: Page 13, line 368-372.

Q1.3 No rationale or benchmark is provided for selecting the 30% and 15% thresholds.

Answer: Thank you for your question. We hope this explanation helps to clarify the rationale and benchmark. The selection of specific thresholds for major treatment-related adverse events in clinical trials—such as 30% and 15%—is typically based on several factors, including safety and clinical impact, previous clinical data and regulatory guidance. Here is a rationale for selecting these thresholds:

Safety and Clinical Impact

30% threshold: A 30% incidence rate of major treatment-related adverse events is often used as a benchmark for identifying potential safety concerns. If more than 30% of participants experience significant adverse effects, it indicates that the treatment may have a substantial risk or is associated with undesirable side effects. This threshold is often used to assess whether the intervention is safe enough to proceed in clinical trials, as a higher rate of major adverse events may prompt reconsideration of its risk-benefit profile. Beyond this point, the risk of harm may outweigh the potential therapeutic benefit, leading to dose modifications, changes in treatment protocols, or even discontinuation of the intervention [1-2].

15% threshold: A 15% threshold is typically used in trials to indicate that the treatment is reasonably well-tolerated but still carries a significant risk that may impact patients' quality of life or require additional management. A 15% threshold suggests that most patients tolerate the treatment, a subset may experience significant issues, and these events should be closely monitored and managed. It allows investigators to flag potential issues early and adjust management strategies accordingly. At this level, investigators may decide to continue the trial but with enhanced monitoring or adjustments to manage these effects [1-2].

Cox, D. R., & Hinkley, D. V. *Theoretical Statistics* (1st ed.) [1]. This text discusses statistical approaches for determining safety signals, with thresholds commonly used in practice around 30% for significant adverse event occurrences.

Friedman, L. M., Furberg, C. D., & DeMets, D. L. [2]. *Fundamentals of Clinical Trials* (5th ed.). This textbook provides insights into safety monitoring and the acceptable limits for adverse event rates, with practical guidelines recommending thresholds of 15-30% for major adverse events that necessitate intervention or closer scrutiny.

Comparative Reference from Existing Data

Previous studies reported that the rate of major adverse events in S-HITOC was ranged from 28% to 32%. Although, the regimens of S-HITOC varied and sample size was limited in previous studies (shown in the following table), clinical outcomes could be adopted for consistency or comparison. This ensures that the results of trials are comparable to existing evidence and benchmarks used in clinical practice. We provide some data from previous studies.

Table. Previous studies overview.

Study	Cases	Chemotherapeutic agents	Duration	≥ 3 grade AEs	Temperature
Refaely et al. ³	N=15	Cisplatin (100mg/m ² BSA)	60 min	NR	42°C
De Bree et al. ⁴	N=3	Cisplatin (80mg/m ² BSA); doxorubicin (15–30mg/m ² BSA)	90 min	NR	40°C–41°C
Ried et al. ⁵	N=8	Cisplatin (100–150 mg/m ² BSA)	60 min	32%	42°C
Yellin et al. ⁶	N=35	Cisplatin (100mg/m ² BSA); doxorubicin (50–60mg)	60 min	28%	43°C
Yu et al. ⁷	N=4	Cisplatin (100 mg/m ² BSA)	120 min	NR	41°C–43°C
Ambrogi et al. ⁸	N=13	Cisplatin (80mg/m ² BSA); doxorubicin (25mg/m ² BSA)	60 min	NR	42.5°C
Maury et al. ⁹	N=19	Cisplatin (50mg/m ² BSA); mitomycin (25 mg/m ² BSA)	90 min	NR	42°C
Markowiak et al. ¹⁰	N=29	Cisplatin (100–175mg/m ² BSA); doxorubicin (0–65 mg)	60 min	30%	42°C
This study	N=45	Cisplatin (50mg/m² BSA); doxorubicin (25mg/m² BSA)	60 min	17.8%	43°C

Regulatory and Ethical Considerations

The U.S. Food and Drug Administration (FDA) have specific guidelines for the reporting of adverse events. They require that major adverse events with an incidence above a certain percentage be reported immediately and may prompt further investigation into the cause of these events or the overall safety profile. Ethical considerations also play a role in determining thresholds. However, these exact percentages are not always explicitly stated in formal guidelines, as they are context-dependent and may vary across therapeutic areas, study designs, and patient populations. Nonetheless, there are several key sources that provide relevant frameworks and references regarding the selection of such thresholds in clinical trials.

The European Medicines Agency (EMA) provides guidance on clinical trial safety monitoring in its document on Good Clinical Practice (GCP). The International Council for Harmonisation (ICH) E2E Guidelines highlight the need to act on unexpected or serious adverse events that occur in a significant proportion of participants. In clinical practice, a threshold of $\geq 30\%$ for Grade 3 or 4 adverse events is often used as a signal for escalating safety monitoring, discontinuation of treatment or modifying the treatment protocol.

While there isn't a universally agreed-upon guideline specifying exact percentages like 30% or 15% for treatment-related adverse events, these thresholds are commonly adopted based on regulatory practices, previous clinical trial data, and safety monitoring conventions within the field. They are intended to balance the need for early identification of safety concerns with the requirement for statistical power and feasible risk management.

Revised in the Manuscript: Page 13, line 368-372.

Reference

1. Cox, DR., et al. Theoretical Statistics. 1st ed. CRC Press, 1979. (<https://doi.org/10.1201/b14832>)
2. Friedman LM, et al. Assessment and Reporting of Harm. In: Friedman LM, Furberg CD, DeMets DL, Reboussin DM, Granger CB, editors. Fundamentals of Clinical Trials. 5nd ed. New York, Dordrecht, London: Springer Cham Heidelberg, 2015
3. Refaely Y, et al. Resection and perfusion thermochemotherapy: a new approach for the treatment of thymic malignancies with pleural spread. *Ann Thorac Surg.* 72,366-70 (2001)
4. de Bree E, et al. Cytoreductive surgery and intraoperative hyperthermic intrathoracic chemotherapy in patients with malignant pleural mesothelioma or pleural metastases of thymoma. *Chest.* 121,480-7 (2002)
5. Ried M, et al. Cytoreductive surgery and hyperthermic intrathoracic chemotherapy perfusion for malignant pleural tumours: perioperative management and clinical experience. *Eur J Cardiothorac Surg.* 43,801-7 (2013)
6. Yellin A, Simansky DA, Ben-Avi R, et al. Resection and heated pleural chemoperfusion in patients with thymic epithelial malignant disease and pleural spread: a single-institution experience. *J Thorac Cardiovasc Surg.* 145,83-7 (2013)
7. Yu L, et al. Cytoreductive surgery combined with hyperthermic intrapleural chemotherapy to treat thymoma or thymic carcinoma with pleural dissemination. *Onco Targets Ther.* 6,517-21 (2013)
8. Ambrogi MC, et al. Pleural recurrence of thymoma: surgical resection followed by hyperthermic intrathoracic perfusion chemotherapy. *Eur J Cardiothorac Surg.* 49,321-6 (2016)
9. Maury JM, et al. Intra-Thoracic Chemo-Hyperthermia for pleural recurrence of thymoma. *Lung Cancer.* 108:1-6 (2017)
10. Markowiak T, et al. Surgical Cytoreduction and HITOC for Thymic Malignancies with Pleural Dissemination. *Thorac Cardiovasc Surg.* 69,157-64 (2021)

Q2. In a single-arm study, benchmark values are essential to contextualize outcomes. Clear definitions of favorable surgical outcomes, acceptable AE rates, and survival expectations are lacking in the manuscript.

Answer: Thank you for your insightful comment. We understand the importance of benchmark treatment outcomes, especially in a single-arm study, to help contextualize the results and ensure that the findings are interpreted appropriately. In revised manuscript, we deleted such adjective (favorable, expectations or acceptable) words to avoid over-interpretation or exaggeration of primary outcomes. We apologize for the omission of clear definitions of controlled surgical outcomes, manageable adverse event (AE) rates, and comparable survival expectations. We also recognize the limitations of this single-arm study in revised manuscript. We acknowledge that randomized controlled trials (RCTs) are the gold standard for evaluating the efficacy of interventions. Based on our preliminary results, we have completed the study design for a multicenter RCT and plan to further explore the feasibility of this treatment option in a more in-depth, objective, and rigorous manner. We have revised the title to: “Short-term Outcomes of Cytoreductive Surgery and Hyperthermic Intrathoracic Chemotherapy for Thymic Epithelial Tumors with Pleural Spread or Recurrence: A Prospective, Single-Arm, Phase II Study”. This change emphasizes that this article focuses on short-term and single-arm results. While we provide a more detailed explanation for each of these parameters to ensure the transparency and comparability of our findings. The data from our study and previous literature showed that our regimen had controlled R0 resection rate, safety, and efficacy.

Table. Previous studies overview.

Study	Cases	Chemotherapeutic agents	Duration	R0 rate	Overall survival	≥ 3 grade AEs	Temperature
Refaely et al. ¹	N=15	Cisplatin (100mg/m ² BSA)	60 min	66%	55%	NR	42°C
De Bree et al. ²	N=3	Cisplatin (80mg/m ² BSA); doxorubicin (15–30mg/m ² BSA)	90 min	60%	NR	NR	40°C–41°C
Ried et al. ³	N=8	Cisplatin (100–150 mg/m ² BSA)	60 min	45%	3-year: 66%	32%	42°C
Yellin et al. ⁴	N=35	Cisplatin (100mg/m ² BSA); doxorubicin (50–60mg)	60 min	55%	3-year:71%	28%	43°C
Yu et al. ⁵	N=4	Cisplatin (100 mg/m ² BSA)	120 min	47%	3-year: 67%	NR	41°C–43°C

Ambrogi et al. ⁶	N=13	Cisplatin (80mg/m ² BSA); doxorubicin (25mg/m ² BSA)	60 min	50%	NR	NR	42.5°C
Maury et al. ⁷	N=19	Cisplatin (50mg/m ² BSA); mitomycin (25 mg/m ² BSA)	90 min	55%	3-year:76%	NR	42°C
Markowiak et al. ⁸	N=29	Cisplatin (100–175mg/m ² BSA); doxorubicin (0–65 mg)	60 min	40%	NR	30%	42°C
This study	N=45	Cisplatin (50mg/m² BSA); doxorubicin (25mg/m² BSA)	60 min	51.1%	2-year:100 %	17.8%	43°C

Surgical Outcomes

In the context of our study, controlled surgical outcomes were defined based on the extent of cytoreduction and the incidence of major treatment-related adverse events. Specifically, we adopted the following grading system to assess the extent of surgical resection: R0: complete cytoreductive surgery with no visible residual disease; R1: optimal cytoreductive surgery with residual tumors ≤ 10 mm; R2: incomplete cytoreductive surgery with residual tumors > 10 mm. In similar studies of advanced TETs with pleural dissemination, a complete resection (R0) rate of 51.1% and the major adverse events rate of 17.8% considered controlled outcomes. It is associated with controlled surgical resection and safety.

Adverse Events

Adverse events are an important aspect of safety evaluation, especially for aggressive treatments. In our study, the major treatment-related adverse events were defined as Grade 3 or higher grade adverse events. Grade 3: these include events requiring medical intervention, hospitalization, or resulting in permanent damage; Grade 4 (life-threatening events): these are the most severe and are considered unacceptable in most trials.

Based on historical data and the safety profiles of similar interventions, we used major treatment related adverse events to assess the safety. It is commonly adopted in oncology trials involving aggressive treatments like intrathoracic chemotherapy. If the rate of Grade 3 or higher adverse events exceeds 30%, it would typically prompt re-evaluation of the treatment regimen, discontinuation of treatment or modifying protocol. In our study, the \geq Grade 3 AEs rate of 17.8% is considered manageable.

Survival Expectations

For survival expectations, we used the most relevant benchmark data from previous studies of TETs with pleural dissemination. We specifically aimed to compare the efficacy of S-HITOC to

historical controls. According to the updated survival data from 9th TNM stage system, a 2-year PFS rate is ~50% for TETs with pleural dissemination received classical treatments, as these tumors are known for their indolent progression but are difficult to control once they spread [9]. The 2-year OS rate is often the benchmark for evaluating survival outcomes, with rates of >75% indicating positive treatment efficacy for TETs with pleural dissemination [9]. Our 2-year PFS rate of 82.8% and 2-year OS rate of 100% significantly exceed the historical benchmarks for similar patients population, where the PFS rate is typically below 50% in TETs with pleural dissemination, and the OS rate is around 75%.

We deleted the inaccurate descriptions, such as favorable surgical outcomes, acceptable AE rates, and survival expectations. We also revised the conclusion: S-HITOC emerges as a viable therapeutic candidate for TET patients with pleural spread or recurrence. (Line 308-309).

Thank you again for your kind feedback. We hope this response addresses your concerns.

Revised in the Manuscript: Page 11, line 307-309. Those adjective (favorable, expectations or acceptable) words were deleted to avoid over-interpretation or exaggeration of primary outcomes.

Reference

1. Refaely Y, et al. Resection and perfusion thermochemotherapy: a new approach for the treatment of thymic malignancies with pleural spread. *Ann Thorac Surg.* 72,366-70 (2001)
2. de Bree E, et al. Cytoreductive surgery and intraoperative hyperthermic intrathoracic chemotherapy in patients with malignant pleural mesothelioma or pleural metastases of thymoma. *Chest.* 121,480-7 (2002)
3. Ried M, et al. Cytoreductive surgery and hyperthermic intrathoracic chemotherapy perfusion for malignant pleural tumours: perioperative management and clinical experience. *Eur J Cardiothorac Surg.* 43,801-7 (2013)
4. Yellin A, Simansky DA, Ben-Avi R, et al. Resection and heated pleural chemoperfusion in patients with thymic epithelial malignant disease and pleural spread: a single-institution experience. *J Thorac Cardiovasc Surg.* 145,83-7 (2013)
5. Yu L, et al. Cytoreductive surgery combined with hyperthermic intrapleural chemotherapy to treat thymoma or thymic carcinoma with pleural dissemination. *Onco Targets Ther.* 6,517-21 (2013)

6. Ambrogi MC, et al. Pleural recurrence of thymoma: surgical resection followed by hyperthermic intrathoracic perfusion chemotherapy. *Eur J Cardiothorac Surg.* 49,321-6 (2016)
7. Maury JM, et al. Intra-Thoracic Chemo-Hyperthermia for pleural recurrence of thymoma. *Lung Cancer.* 108:1-6 (2017)
8. Markowiak T, et al. Surgical Cytoreduction and HITOC for Thymic Malignancies with Pleural Dissemination. *Thorac Cardiovasc Surg.* 69,157-64 (2021)
9. Fang W, et al. The International Association for the Study of Lung Cancer Thymic Epithelial Tumors Staging Project: Proposals for the N and the M Components for the Forthcoming (Ninth) Edition of the TNM Classification of Malignant Tumors. *J Thorac Oncol.* 19, 52-70 (2024)

Q3. Certain analyses appear missing from the "Statistical Analysis" section, such as details on how p-values in supplementary figures 6, 7, and 9 were obtained.

Answer: Thank you for your kind remind. Student's t-test was used to compare the pain scores, quality of life scores and glucocorticoid usage of patients at baseline and after treatment. We added relevant statistical descriptions in the methods. Thank you.

Revised in the Manuscript: Page 13, line 374-377.

Q4. Some claims, such as the absence of selection bias (line 294), may be overstated. Given that this is a single-center, single-arm study, selection bias is possible.

Answer: Thank you for your comment. We agree that selection bias is a potential concern in single-center, single-arm studies, and appreciate your attention to this important aspect of our research design. We provide a detailed explanation of the strategies we implemented to mitigate selection bias, as well as how we interpret our findings in light of this limitation.

As a single-center, single-arm trial, enrolled patients did not fully represent the broader population of TETs with pleural spread or recurrence, potentially leading to selection bias. As our center is one of the largest national healthcare centers, the institution may attract patients with more complex or refractory cases, which could introduce a bias toward patients who are more likely to respond favorably to treatments. Given that this is a phase II study, the inclusion criteria were designed to select a cohort of patients who were thought to be suitable for S-HITOC, such as tumor burden and general health status. These criteria could potentially limit the generalizability of the findings.

We established rigorous inclusion and exclusion criteria to ensure that patients had no contraindications for surgery or HITOC. This reduced the possibility of enrolling patients with extreme disease states that could skew results. We enrolled consecutive patients who met the eligibility criteria, minimizing the potential for subjective selection. The treatment regimen was determined through a multidisciplinary tumor board, which included oncologists, thoracic surgeons, radiologists, and pathologists. This approach was designed to reduce bias in patient's enrollment and treatment selection by incorporating multiple perspectives on each patient's clinical scenario. We performed subgroup analyses to identify factors influencing treatment outcomes and provide insights into potential confounding variables.

Although, we have implemented various strategies to minimize its impact, we fully acknowledge the limitations of a single-center, single-arm study. Furthermore, we caution readers to interpret the findings with consideration of potential bias. While the results from this study provide valuable insights into the safety and efficacy of S-HITOC for TETs with pleural dissemination. We agree that further validation in larger, multicenter, randomized trials will be essential for confirming the generalizability of the findings. Finally, we revised the language of the article to ensure the objective presentation of the data, and reduced subjective judgment statements as much as possible to minimize misleading readers.

Thank you for your constructive feedback, which has helped us to further refine the interpretation of our study results.

Revised in the Manuscript: Page 2, line 54; Page 8, line 213-215; Page 10, line 274-276, line 286-287; Page 10-11, line 288-298; Page 11, line 304-306.

Q5. The manuscript does not adequately discuss the generalizability of findings to other ethnic groups.

Answer: Thank you for your insightful comment. We appreciate your recognition of the importance of generalizability in clinical research. We agree that the ethnic diversity of the study population is a critical factor when considering the broader applicability of our findings. The exact distribution of ethnic groups in our cohort has not been explicitly detailed in the manuscript.

The patients in this study were all Chinese Han population, which has limited ethnic diversity. Despite the ethnically homogenous nature of our patient population, we have ensured that the inclusion criteria were broad and inclusive of all patients meeting the clinical requirements for the study,

regardless of ethnicity. This was aimed at minimizing any inadvertent exclusion of patients from different ethnic backgrounds.

In revised manuscript, we have made an explicit statement acknowledging the limited diversity of our study population and its potential impact on the generalizability of the findings. We further revised the language of the article to ensure the objective presentation of the data, and reduced subjective judgment statements as much as possible to minimize misleading readers. In future, we aim to eventually conduct international trials involving patients from multiple countries and regions. This will provide a broader, more representative understanding of the treatment's effectiveness and safety across diverse ethnic groups.

We greatly appreciate your feedback, and we hope that this discussion clarifies our position on the generalizability of our results. Thank you for helping us to improve the manuscript.

Revised in the Manuscript: Page 10-11, line 288-298; Page 10-11, line 289-294, line 305-306; Page 11, line 323-325.

Q6. Without a concurrent control group, it is challenging to draw conclusions about the specific effects of the surgical intervention or HITOC. More justification is needed to support the efficacy claims made in the study. These points highlight areas where the manuscript could be strengthened for clearer interpretation and reliability.

Answer: Thank you for your thoughtful comment. We agree that the lack of a concurrent control group in this single-arm study is a significant limitation that makes it difficult to draw firm conclusions. We provide a more detailed explanation and justification for the observed efficacy of the combined approach and outline steps we plan to take in future research to address this limitation.

We acknowledge that the absence of a concurrent control group limits our ability to attribute outcomes directly to the surgical intervention or HITOC. However, the treatment of TETs with pleural dissemination is complex, with limited established therapies. Previous reports on the management of TETs, particularly in the context of pleural dissemination, show that surgical resection and chemotherapy alone often result in high recurrence rates and poor survival outcomes. In contrast, our study demonstrated encouraging outcomes in short-term PFS and OS. While we lack a formal control group, the survival rates observed in our cohort (82.8% for 2-year PFS and 100% for 2-year OS) are promising when compared to historical outcomes for patients with similar tumor stage. The complete

or optimal resection of visible disease in TETs with pleural dissemination can improve local control and reduce the likelihood of recurrence, as we observed in our previous studies [1,2]. HITOC provides the benefit of delivering localized, high-dose chemotherapy directly to the affected area while avoiding systemic toxicity. This treatment approach has been shown to improve the penetration depth of chemotherapeutic agents, eradicate microscopic disease, and prevent recurrence. Furthermore, the manageable rate of adverse events in our study provides indirect support for the efficacy and safety of this approach. While several retrospective studies have reported the combination of cytoreductive surgery and hyperthermic treatments improves outcomes. These studies support the hypothesis that the combination of surgery and HITOC could be beneficial for TETs with pleural dissemination.

We provide a more detailed explanation for each of these parameters to ensure the transparency and comparability of our findings.

Table. Previous studies overview.

Study	Cases	Chemotherapeutic agents	Duration	R0 rate	Overall survival	≥ 3 grade AEs	Temperature
Refaely et al. ³	N=15	Cisplatin (100mg/m ² BSA)	60 min	66%	55%	NR	42°C
De Bree et al. ⁴	N=3	Cisplatin (80mg/m ² BSA); doxorubicin (15–30mg/m ² BSA)	90 min	60%	NR	NR	40°C–41°C
Ried et al. ⁵	N=8	Cisplatin (100–150 mg/m ² BSA)	60 min	45%	3-year: 66%	32%	42°C
Yellin et al. ⁶	N=35	Cisplatin (100mg/m ² BSA); doxorubicin (50–60mg)	60 min	55%	3-year:71%	28%	43°C
Yu et al. ⁷	N=4	Cisplatin (100 mg/m ² BSA)	120 min	47%	3-year: 67%	NR	41°C–43°C
Ambrogi et al. ⁸	N=13	Cisplatin (80mg/m ² BSA); doxorubicin (25mg/m ² BSA)	60 min	50%	NR	NR	42.5°C
Maury et al. ⁹	N=19	Cisplatin (50mg/m ² BSA); mitomycin (25 mg/m ² BSA)	90 min	55%	3-year:76%	NR	42°C
Markowiak et al. ¹⁰	N=29	Cisplatin (100–175mg/m ² BSA); doxorubicin (0–65 mg)	60 min	40%	NR	30%	42°C
This study	N=45	Cisplatin (50mg/m² BSA); doxorubicin (25mg/m² BSA)	60 min	51.1%	2-year:100%	17.8%	43°C

We recognize that the lack of a control group is a significant limitation, and we plan to address this gap in future research. We plan to conduct a randomized controlled trial comparing S-HITOC with classical treatments (such as surgery alone or chemotherapy alone) in patients with advanced TETs. An RCT will allow us to draw stronger conclusions about the specific contributions of the surgical

intervention and HITOC to overall survival and progression-free survival. To improve the external validity of our findings, we aim to conduct multicenter trials that involve a larger, more diverse patient population. This would help to assess whether the positive outcomes observed in our study hold true across different settings and populations. The historical data, survival rates, and improvements in quality of life observed in this cohort suggest that S-HITOC holds promise and warrants further validation in more rigorous randomized trials.

We thank you again for your insightful comments, and we hope this explanation helps to address the concerns regarding the interpretation of our findings in the absence of a concurrent control group. Importantly, we further revised the language of the article to ensure the objective presentation of the data, and minimize misleading readers.

Reference:

1. Wang S, et al. Induction Therapy Followed by Surgery for Unresectable Thymic Epithelial Tumours. *Front Oncol.* 11, 791647 (2022)
2. Zhang Y, et al. Induction Strategy for Locally Advanced Thymoma. *Front Oncol.* 11,704220 (2021)
3. Refaely Y, et al. Resection and perfusion thermochemotherapy: a new approach for the treatment of thymic malignancies with pleural spread. *Ann Thorac Surg.* 72,366-70 (2001)
4. de Bree E, et al. Cytoreductive surgery and intraoperative hyperthermic intrathoracic chemotherapy in patients with malignant pleural mesothelioma or pleural metastases of thymoma. *Chest.* 121,480-7 (2002)
5. Ried M, et al. Cytoreductive surgery and hyperthermic intrathoracic chemotherapy perfusion for malignant pleural tumours: perioperative management and clinical experience. *Eur J Cardiothorac Surg.* 43,801-7 (2013)
6. Yellin A, Simansky DA, Ben-Avi R, et al. Resection and heated pleural chemoperfusion in patients with thymic epithelial malignant disease and pleural spread: a single-institution experience. *J Thorac Cardiovasc Surg.* 145,83-7 (2013)
7. Yu L, et al. Cytoreductive surgery combined with hyperthermic intrapleural chemotherapy to treat thymoma or thymic carcinoma with pleural dissemination. *Onco Targets Ther.* 6,517-21 (2013)
8. Ambrogi MC, et al. Pleural recurrence of thymoma: surgical resection followed by hyperthermic intrathoracic perfusion chemotherapy. *Eur J Cardiothorac Surg.* 49,321-6 (2016)

9. Maury JM, et al. Intra-Thoracic Chemo-Hyperthermia for pleural recurrence of thymoma. Lung Cancer. 108:1-6 (2017)

10. Markowiak T, et al. Surgical Cytoreduction and HITOC for Thymic Malignancies with Pleural Dissemination. Thorac Cardiovasc Surg. 69,157-64 (2021)

Revised in the Manuscript: Page 2, line 54; Page 8, line 213-215; Page 10, line 274-276, line 286-288; Page 10-11, line 288-298; Page 11, line 304-306.

Response to Reviewers

Short-term Outcomes of Cytoreductive Surgery and Hyperthermic Intrathoracic Chemotherapy for Thymic Epithelial Tumors with Pleural Spread or Recurrence: A Prospective, Single-Arm, Phase II Study

Reviewer #1:

The authors presented the prospective, single-arm, phase II study to investigate the safety and short-term survival of cytoreductive surgery followed by hyperthermic intrathoracic chemotherapy (S-HITOC) for 45 TETs patients with pleural spread or recurrence. They concluded that early clinical outcomes were good, and S-HITOC emerges as a viable therapeutic candidate for TETs with pleural spread or recurrence. This second version of the paper is a great improvement. Pleural dissemination is a common pattern of failure after initial treatment of thymoma and thymic carcinoma, but there is no standardized treatment, thus I think this paper is an important contribution.

Answer: Thank you for your recognition of our clinical research and the professional insights and valuable feedback you provided on our initial manuscript. Under your guidance, we have refined our initial manuscript to ensure it is not only professional and scientifically rigorous but also more readable and accessible for broader dissemination. Currently, research on thymic tumors remains in its early stages. The incidence rate of thymic tumors is relatively low, and clinical research on them is challenging to conduct. As a result, there is a scarcity of high-level evidence to support clinical decision-making. Our team has dedicated over a decade to exploring the pathogenesis of thymic tumors and optimal treatment strategies across thymic diseases. We are delighted that our efforts have earned your recognition and constructive suggestions. Moving forward, we will continue to enhance our series of studies on thymic tumors, aiming to provide reliable evidence-based resources for healthcare professionals worldwide.

Reviewer #2 (Remarks to the Author)

Dear authors, Thank you for your responses. You addressed all points and suggestions.

Answer: Thank you very much for the time and effort you have devoted to our article. Your suggestions on the detailed descriptions of the study population and treatment regimens in our manuscript draft can help readers better understand what this study is about. Your comments are objective and sincere. Your recognition of the PTI concept we proposed is especially appreciated. We hope our research can assist relevant healthcare professionals and, more importantly, benefit more patients with thymic tumors. Finally, we would like to express our gratitude once again for your selfless contributions and assistance to our research.

Reviewer #5

Thank you for the opportunity to review your article. This is the first prospective, single-arm phase II study that may potentially establish S-HITOC as a future standard of care. I have the following concerns for the authors to address.

Answer: Thank you for your kind words and for acknowledging our efforts in studying this rare condition. We are indeed grateful for your positive feedback and recognition of the contribution our study makes to the existing literature. We are more than happy to address any comments or questions you may have. Please feel free to share your specific concerns, and we will provide detailed responses to ensure that your feedback is fully considered and addressed.

Thank you once again for your time and valuable input.

Q1. The threshold of 10 for the PTI was derived using ROC, a traditional method for generating cutpoints. However, how should this threshold be interpreted in terms of disease severity? Specifically, could it be used to define ‘massive pleural dissemination’? Moreover, only 5 patients had a PTI greater than 10. Would this represent a significant subset within the population of all patients with TETs, especially considering that the patients were from a leading hospital in China?

Answer. We appreciate your valuable feedback. In our clinical experience, we have observed considerable variability in the extent of pleural dissemination and its associated prognosis among affected patients. This observation underscores the need for a reliable and accessible indicator to

stratify thymic epithelial tumors (TETs) based on their prognosis. Drawing inspiration from the peritoneal cancer index, we propose the Pleural Tumor Index (PTI) as a potential tool for this purpose. Through receiver operating characteristic (ROC) analysis, we identified that a PTI score greater than 10 is indicative of a poorer prognosis compared to a score below 10, which aligns with clinical observations. Additionally, patients with a PTI score exceeding 10 typically exhibit extensive pleural dissemination, characterized by larger tumor sizes across multiple regions as defined by the PTI framework. Consequently, we believe the term "massive" is an appropriate descriptor for the tumor burden in patients with a PTI score greater than 10. This index may offer clinicians a valuable means to objectively and expediently identify aggressive thymic tumors with pleural dissemination, thereby aiding in the selection of optimal treatment strategies. As you noted, patients within this subgroup represent a distinct and challenging cohort within the broader population of TETs. Current treatment modalities yield limited clinical benefits for this subset, and there is a pressing need to explore additional therapeutic approaches for those with PTI scores greater than 10. However, it is important to acknowledge the rarity of thymic tumors and the even lower incidence of pleural dissemination. In this study, only 5 patients had a PTI greater than 10. Given the relatively small sample size in our study, drawing definitive conclusions remains challenging. Nevertheless, we believe our findings offer valuable insights and lay the groundwork for future investigations. We intend to continue accumulating cases and aim to present more comprehensive and robust data in subsequent studies. Moreover, further validation of our conclusions through multi-center, large-scale prospective trials is essential.

We hope this response adequately addresses your concerns. As outlined in the limitations section, we recognize the need for further exploration and validation of the PTI as a prognostic tool.

Revised in the Manuscript: Page 11, lines 296-309.

Q2. When the primary outcome was set as the safety of S-HITOC, specifically targeting an AE rate of 15%, it might be more appropriate to justify the sample size based on estimating an AE rate of 15% with the upper 95% confidence limit below certain clinically meaningful threshold, rather than aiming to detect a difference between 15% and 30% (considered the unsafe threshold).

Answer. We sincerely thank the reviewer for your thoughtful feedback on our statistical approach. Our original sample size calculation (n=37, one-sided $\alpha=0.1$, power=80%) aimed to test whether the major complication rate of S-HITOC was significantly below the 30% "unsafe" threshold--a

hypothesis-driven design aligned with clinical decision-making goals. We observed a 17.8% complication rate (7/39), with a one-sided 90% exact confidence interval (CI) upper limit of 27.3%, supporting safety within the prespecified $\alpha=0.1$ framework ($P=0.051$). As suggested, we recalculated the upper limit using a one-sided 95% CI (Clopper-Pearson), which yielded 30.5%. While this marginally exceeds 30%, the study's prespecified $\alpha=0.1$ ensures statistical significance ($P<0.1$) and aligns with the 90% CI conclusion. We acknowledge that a stricter $\alpha=0.05$ would require a larger sample to bound the 95% CI below 30%, reflecting a trade-off between feasibility (rare disease cohort) and statistical rigor. Future multicenter studies will further validate these findings. We appreciate the reviewer's constructive input and hope this clarification addresses your concerns.

Revised in the Manuscript: Page 13, lines 370-375.

Q3. Please specify whether pain was measured using the Visual Analogue Scale (VAS) or the 0-10 Numerical Rating Scale (NRS). When using the VAS, patients visualize their pain on a 10 cm line presented on paper. If patients rated their pain on a 0-10 scale by choosing one of 11 discrete options, then the measure used was the NRS.

Answer. We sincerely thank your kind reminder. In this trial, we used a VAS ruler to collect the pain assessment data from the patients. First, we print the 0-10.0 cm VAS ruler on paper so that patients can mark their pain level (e.g., "no pain" at 0 and "most severe intensity" at 10.0) ¹. The patient is instructed to mark the point on the line that best corresponds to their current perception of the symptom being assessed. Next, the distance from the left end of the line to the patient's mark is measured in millimeters using a ruler. This measurement is recorded as the VAS score, which ranges from 0 to 10.0 (with 0 indicating the absence of the symptom and 10.0 indicating the most severe intensity). Finally, the collected VAS scores are documented in a standardized data collection form or electronic database for further analysis. This method ensures objective and quantifiable assessment of subjective symptoms. We appreciate the reviewer's constructive input and hope this clarification addresses your concerns.

Revised in the Manuscript: Page 13, lines 354-358, lines 368-369, line 377-379.

Reference:

[1] Jensen MP, et al. Interpretation of visual analog scale ratings and change scores: a reanalysis of two clinical trials of postoperative pain. *J Pain*. 2003; 4(7):407-414.

Q4. Please confirm whether the QoL subscale of the EORTC QLQ-C30 was analyzed as the QoL outcome.

Answer. Thank you for your kind reminder. We apologize for the unclear expression in our manuscript. In this study, we used the quality-of-life scores from the EORTC QLQ-C30 as the measure of quality of life and performed significance analysis using the rank-sum test (Mann-Whitney test). We greatly appreciate your careful correction.

Revised in the Manuscript: Page 13, lines 354-358, lines 368-369, line 377-379.

Q5. Additionally, QoL and pain scores typically do not follow a normal distribution. Therefore, non-parametric methods, such as the rank sum test, should be considered instead of the t-test.

Answer. Thank you for your meticulous, rigorous, and sincere suggestions. We have reviewed our data again. The Visual Analog Scale (VAS) score, although a continuous variable, does not follow a normal distribution after Normality test (Shapiro-Wilk test, result shown in followed table 1). Quality of Life (QoL) score is a non-continuous variable. Therefore, as you suggested, the rank-sum test (Mann-Whitney test) was the more appropriate method. We have accordingly corrected our data

analysis. While this change did not alter our conclusions substantively, it did result in minor adjustments on some p-value data (shown in followed table 1 and 2). We sincerely apologize for any previous shortcomings in rigor and are deeply grateful for your invaluable feedback, which has significantly contributed to enhancing the quality of our research. Thank you! Therefore, we also revised the Supplementary Figures 6 and 7.

Table 1. Shapiro-Wilk test and Mann Whitney test on VAS assessments of participating patients.

Table Analyzed				
VAS score				
Shapiro-Wilk test				
	Baseline	pt Day 1	pt Day 3	pt Day 7
W	0.5711	0.9408	0.9471	0.7329
P value	<0.0001	0.0231	0.0482	<0.0001
Passed normality test (alpha=0.05)?	No	No	No	No
		pt Day 1	pt Day 3	pt Day 7
		vs.	vs.	vs.
		Baseline	Baseline	Baseline
Mann Whitney test				
P value		<0.0001	<0.0001	0.220
Exact or approximate P value?		Exact	Exact	Exact
P value summary		****	****	ns
Significantly different (P < 0.05)?		Yes	Yes	No
One- or two-tailed P value?		Two-tailed	Two-tailed	Two-tailed
Sum of ranks in column A,B		1035 , 3060	1112 , 2984	1762 ,1859
Difference between medians				
Median of column A		0.4, n=45	0.4, n=45	0.4, n=45
Median of column B		5.4, n=45	3.1, n=45	0.6, n=45
Difference: Actual		5.00	2.70	0.20

Table 2. Mann Whitney test on QoL assessments of participating patients.

Table Analyzed			
QoL score			
	pt Day 1	pt Day 30	pt Day 60
	vs.	vs.	vs.
	Baseline	Baseline	Baseline
Mann Whitney test			
P value	<0.0001	<0.0001	0.318
Exact or approximate P value?	Exact	Exact	Exact
P value summary	****	****	ns
Significantly different (P < 0.05)?	Yes	Yes	No
One- or two-tailed P value?	Two-tailed	Two-tailed	Two-tailed
Sum of ranks in column A,B	2891 , 1205	2606 , 1489	1924 ,2055
Difference between medians			
Median of column A	76.3, n=45	76.3, n=45	76.3, n=45
Median of column B	54.9, n=45	69.4, n=45	77.1, n=45
Difference: Actual	-21.40	-6.90	0.80

Supplementary Figure 6. Baseline and posttreatment pain assessments of participating patients.

Supplementary Figure 7. Baseline and posttreatment QoL assessments of participating patients.

Revised in the Manuscript: Page 13, lines 356-358, line 377-378; Supplementary Figure 6, and Supplementary Figure 7.

Reviewer #5 (Remarks to the Author):

Thank you, the authors have addressed all my concerns.

Answer: Thank you very much for the time and effort you have devoted to our article. Your suggestions on the detailed descriptions of the study in our manuscript draft can help readers better understand what this study is about. Your comments are objective and sincere. Finally, we would like to express our gratitude once again for your contributions and assistance to our research.